# XoRA: Expander Adapted LoRA Finetuning

## Abstract

Parameter-efficient fine-tuning aims to reduce the computational cost of adapting foundational models to downstream tasks. Low-rank matrix based adaptation (LoRA) techniques are popular for this purpose. We propose XoRA, an efficient fine-tuning scheme, which sparsifies the low-rank matrices even further using expander masks. The mask is generated using extremal expander graphs (Ramanujan graphs) to maintain high edge connectivity even at a very high sparsity. Experimental results demonstrate that this method has comparable performance with the LoRA fine-tuning method while retaining much fewer number of parameters.

## 1 Introduction

Large language models are often fine-tuned for improving their performance on downstream tasks. Computational and memory requirement of such retraining is reduced by using parameter-efficient fine-tuning (PEFT) (Ding et al., 2023b; Lialin et al., 2023; Han et al., 2024). Most popular among them are the reparameterization based techniques, pioneered by the Low-Rank Adaptation (LoRA) algorithm (Hu et al., 2021). It adapts the original set of weights ($W_0$) using a rank constrained decomposition of the weight update ($\Delta W = BA$) matrix into up and down projection matrices $A$ and $B$. Various modifications to LoRA has been recently suggested in literature (Mao et al., 2024).

It has been observed that the LoRA low-rank matrices has a considerable redundancy. They can be sparsified further (Wu et al., 2024) without significant loss of performance. Sparsification of the LoRA up and down projection matrices has been attempted in LoRA-Prune (Zhang et al., 2023b), and Bonsai (Dery et al., 2024). Robust sparse regularizers has been applied during the low-rank matrix decomposition process in RoSA (Nikdan et al., 2024a) to reduce the number of non-zero parameters. The LoTA algorithm (Panda et al., 2024) utilises iterative magnitude pruning to identify sparse winning lottery tickets for the transformers during fine-tuning in LoRA. Random selection of trainable weights have also been shown to be effective for fine-tuning (Xu & Zhang, 2024).

Masking or parameter selection is a popular parameter-efficient fine-tuning method which updates only a subset of the parameters of the original network (Ploner & Akbik, 2024), while keeping the large majority of weights unchanged. This is usually done by applying a binary mask on the weight update matrix. The mask is designed using various criteria like Fisher information (Das et al., 2023), weight magnitudes (Liao et al., 2023), or the change in weight magnitude (Ansell et al., 2021) etc. However, many of the sophisticated weight pruning algorithms are difficult to use for this purpose because of the high computational requirements. Similarly, iterative pruning is time consuming for very large models. Random masks are experimentally found to be less effective at a very high sparsity. This motivates the need for effective structural sparsification algorithms that can be applied on the LoRA low-rank matrices. Expander graphs are sparse but well connected graphs that are useful in designing resilient network structures (Lubotzky, 1994). They have been found to be useful in designing sparse neural networks (Pal et al., 2022; Laenen, 2023) which can be trained to achieve a performance close to that of a dense network.

In this study, we propose an expander graph based structural masking technique on the LoRA projection matrices (XoRA). A block diagram of the proposed approach is shown in Figure 1. We experimentally observe that the LoRA low-rank matrices ($A$ and $B$) can be further sparsified while maintaining the performance. The masking needs to preserve the network connectivity even at a very high sparsity. This can be achieved using a expander graph based mask generation techniques. A significantly higher parameter efficiency is experimentally observed as compared to LoRA.

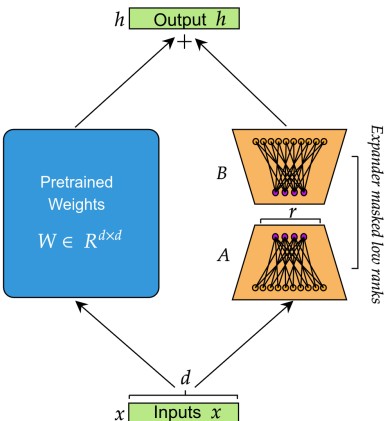

Figure 1: Schematic of the proposed XoRA adaptation algorithm.

## 1.1 CONTRIBUTIONS

The contributions of the article are as follows:

① We propose a methodology for efficiently sparsifying LoRA matrices for finetuning. We achieve comparable or superior performance to LoRA and other state of the art sparsification techniques (see Tables 1 - 9). This also establishes the presence and identification of winning lottery tickets in LoRA matrices.

② It provides a method of reducing the number of parameters in LoRA without reducing the rank further. Related methods like VeRA, which learns scaling vectors, reduces the rank by eliminating entire columns in the matrices.

③ The proposed method may be combined with other adaptation techniques to achieve further parameter efficiency.

④ XoRA demonstrates reduced training time and memory requirements compared to standard LoRA fine-tuning, see section 5.2.4 and Table 7 for details.

## 2 RELATED WORK

Parameter-efficient fine-tuning (PEFT) of transformers has been widely studied in literature (Ding et al., 2023b; Lialin et al., 2023; Han et al., 2024). Major approaches can be categorized as additive, selective, and reparameterized. While additive techniques use additional parameters for fine-tuning to newer tasks, the selective method fine-tune only a subset of the model parameters. Reparameterized methods transform the parameters into equivalent low dimensional forms that are fine-tuned for downstream tasks. Hybrid schemes combine the above approaches.

Low-rank adaptation (LoRA) (Hu et al., 2021) is perhaps the most popular reparameterization based technique. Numerous modifications of LoRA has been suggested in literature (Mao et al., 2024). The strategies include quantization, scaling, and singular value decomposition of the low rank matrices. The VeRA method (Kopiczko et al., 2023) uses a trainable random scaling vector for the shared weights across the layers to achieve a high degree of parameter efficiency. Modifying the low rank matrices by transforming their eigenvectors has been found to be useful for attaining extremely low number of trainable parameters (Bałazy et al., 2024). Spectral adaptation is also used for this purpose (Zhang & Pilanci, 2024). Other modifications of LoRA include (Nikdan et al., 2024b; Ding et al., 2023a; Zhang et al., 2023c) etc.

Selection methods use structured or unstructured masking to determine a subset of the parameters for fine-tuning. The subset is commonly selected using pruning techniques based on the weight magnitude or other information criteria (Liao et al., 2023; Das et al., 2023). Regularization is used during training to obtain a sparse wright distribution in some of these approaches Guo et al. (2021). Structurally selecting some of the parameters like the bias terms also shows benefit for PEFT (Zaken

et al., 2021). Recently, neural architecture search is being employed to find the optimum set of parameters to be selected (Zhou et al., 2024).

Graph structure of the underlying network is analysed by few of the fine-tuning techniques. It has been observed that maintaining connectivity is an important factor in fine-tuning process of a neural network (Liu et al., 2023). Connectivity patterns are found to encode a particular task and may be useful for successful fine-tuning (Xi et al., 2023). Expander graphs have been recently utilized in efficient transformer models. The Diffuser architecture (Feng et al., 2023) uses the expander graph structure to develop sparse attention models over long sequences.

There exist complementary recent finetuning techniques such as IA3, OFT, BOFT, prompt-based methods etc. Input-Aware Parameterization (IA3) (Liu et al., 2022) shifts the focus from weight updates to activation scaling, introducing task-specific learned vectors that scale key, value, and feedforward network activations. Orthogonal Fine-Tuning (OFT) (Qiu et al., 2023) introduces a novel perspective by preserving the hyperspherical energy of pretrained models, ensuring that the semantic consistency of model representations remains intact during fine-tuning. Building upon OFT, Butterfly Orthogonal Fine-Tuning (BOFT) (Liu et al., 2024a) employs sparse butterfly matrix structures inspired by fast Fourier transform algorithms to achieve dense orthogonal transformations with reduced parameter complexity. Complementing these weight-based approaches, prompt-based methods such as in (Liu et al., 2024b; Li & Liang, 2021) etc. leverage learned embeddings concatenated to input sequences or activations. These methods allow task-specific adaptation by modifying the input or internal representations of the model rather than altering its weights, resulting in negligible computational overhead during inference.

## 3 BACKGROUND

### 3.1 LOW-RANK ADAPTATIONS

Low-rank adaptations (LoRA) reduces the number of trainable parameters in large models by injecting low-rank matrices into the model's architecture (Hu et al., 2021). Specifically, it decomposes the weight matrices $W$ into a sum of a frozen pre-trained matrix $W_0$ and a learnable low-rank matrix $\Delta W = BA$.

LoRA defines the weight update for a pre-trained weight matrix $W_0 \in \mathbb{R}^{d \times k}$, as:

$$W = W_0 + \Delta W = W_0 + BA, \tag{1}$$

where $B \in \mathbb{R}^{d \times r}$ and $A \in \mathbb{R}^{r \times k}$ are low-rank matrices, and $r \ll \min(d, k)$ is the rank.

A low-rank matrix $\Delta W \in R^{m \times n}$, with $r \ll \min(m, n)$, can be expressed as $\Delta W = U \Sigma V^\top$, where $U \in R^{m \times r}$, $V \in R^{r \times n}$, and $\Sigma \in R^{r \times r}$ is a diagonal matrix with non-singular values. LoRA is inspired from the studies in Li & Liang (2018) and Aghajanyan et al. (2020) which showed over-parameterized models reside on a low intrinsic dimension. LoRA further hypothesized that the changes in weight $\Delta W$ also has low intrinsic dimension during the model adaptation. Consequently, it uses two learned low-rank matrices $B \in \mathbb{R}^{d \times r}$ and $A \in \mathbb{R}^{r \times k}$ to approximate the weight change $\Delta W$ during adaptation ($\Delta W = BA$). This technique has been exceptionally effective in allowing fine-tuning on low-cost GPU configurations. The optimal dimension $r$ is dependent on the data and determines the number of trainable parameters. Lower the value of $r$ lesser the number of trainable parameters. In our work, XoRA, we experimentally show that the LoRA's low-rank matrices ($B$ and $A$), for a given dimensionality, can be further sparsified while maintaining the performance.

### 3.2 EXPANDER GRAPHS

An expander graph is a sparse graph that has strong connectivity properties, quantified using vertex, edge or spectral expansion. Intuitively, it is is a finite, undirected multigraph in which every subset of the vertices that is not "too large" has a "large" boundary. This can be quantified using the notion of Cheeger constants.

**Definition 3.1** (Expander and Cheeger constant). A graph $\Gamma = (V, E)$ is an $\epsilon$-vertex expander if for every non-empty subset $X \subset V$ with $|X| \leq \frac{|V|}{2}$, we have $\frac{|\delta(X)|}{|X|} \geq \epsilon$, where $\delta(X)$ denotes the outer vertex boundary of $X$ i.e., the set of vertices in $\Gamma$ which are connected to a vertex in $X$ but do not

lie in $X$. As $X$ runs over all subsets of $V$, the infimum of $\frac{|\delta(X)|}{|X|}$ satisfying the conditions above is known as the vertex Cheeger constant and is denoted by $\mathbf{h}_V(\Gamma)$.

Given a graph with a large Cheeger constant, it is difficult to "separate," meaning that it is hard to isolate any subset of vertices from the rest of the graph without cutting many edges. This property facilitates the free flow of information across the entire network and is also known as expansion of a graph and the best expanders are the Ramanujan graphs. Due to space constraints, a detailed review of expanders and Ramanujan graphs is reserved for the appendix, see Appendix A. We shall use Ramanujan graphs to sparsify the LoRA matrices, as can be seen in the next section.

## 4 PROPOSED METHODOLOGY

We first generate bipartite expander graphs with desired number of edges for each of the layers that would be fine-tuned. Their adjacency matrices are then used to mask low-rank weight update matrices ($A$ and $B$) for the corresponding layers of the transformers.

### 4.1 GENERATION OF EXPANDER MASKS

Given an $(n_1, n_2)$ complete bipartite graph, we generate a good expander mask for it. According to the discussion in the previous section, we wish to ensure that this mask has a low degree (in this case $(d_1, d_2)$ bi-degree with $n_1 d_1 = n_2 d_2$ and high Cheeger constant). This brings us to the notion of Ramanujan masks. A Ramaunjan graph is an extremal expander graph in the sense that its spectral gap (and hence also the Cheeger constant) is almost as large as possible. Here, we shall be concerned with bipartite Ramanujan graphs. Recall that a bi-partite graph is said to be balanced if the number of vertices in each of the partitions are the same and it is said to be unbalanced otherwise.

**Definition 4.1** (Bipartite Ramanujan graphs). Let $\Gamma = (V, E)$ be a $d$-regular ($d \geq 3$) balanced bipartite graph. Let the eigenvalues of its adjacency matrix be $\lambda_n \leq \lambda_{n-1} \leq \ldots \leq \lambda_2 \leq \lambda_1$. Then $\Gamma$ is said to be Ramanujan iff $|\lambda_i| \leq 2\sqrt{d-1}$, for $i = 2, \ldots, (n-1)$. For an unbalanced $(d_1, d_2)-$biregular bipartite graph ($d_1, d_2 \geq 3$), the condition of being Ramanujan changes to $|\lambda_i| \leq \sqrt{d_1 - 1} + \sqrt{d_2 - 1}$, for $i = 2, \ldots, (n-1)$.

A detailed description of Ramanujan graphs can be found in (Hoory et al., 2006, sec. 5.3). One can generate the expander (Ramanujan) masks through the following two approaches.

1. Deterministic generation using LPS construction and Ramanujan $r$-coverings.

2. Random generation of bi-regular bipartite graphs and checking for Ramanujan criteria.

To generate expander masks deterministically, we employ the Lubotzky–Phillips–Sarnak (LPS) construction, which produces Ramanujan graphs known for their optimal expansion properties. The LPS method constructs $(p + 1)$-regular graphs using quaternion algebras over number fields, where $p$ is a prime satisfying certain congruence conditions. These graphs exhibit excellent spectral characteristics, making them ideal candidates for our masks. Additionally, Ramanujan $r$-coverings involve creating larger Ramanujan graphs from smaller ones through covering projections while preserving their expansion properties. This approach ensures that the resulting bipartite graphs have the desired bi-degree $(d_1, d_2)$ and high Cheeger constant, which is crucial for the effectiveness of the masked weight matrices in transformers.

Alternatively, the random generation method creates bi-regular bipartite graphs by connecting nodes from two partitions randomly but adhering to the specified degrees $(d_1, d_2)$ and the condition $n_1 d_1 = n_2 d_2$. After constructing such a graph, we compute its adjacency matrix and examine its eigenvalues to check if it meets the Ramanujan criteria—that all non-trivial eigenvalues lie within the bound $|\lambda_i| \leq \sqrt{d_1 - 1} + \sqrt{d_2 - 1}$. This stochastic approach allows for flexibility and ease of implementation, although it may require multiple iterations to find a graph that satisfies the stringent spectral conditions of a Ramanujan graph. Once a suitable graph is found, its adjacency matrix serves as an effective expander mask for the transformer layers.

#### 4.1.1 COMPLEXITY OF MASK GENERATION

Using the notations of the previous section, the computational complexity for generating the Ramanujan graph of is $O(d_1 n_1 n_2)$. For adaptation to language models, when we use the fixed LoRA rank $r$, we can substitute $r = n_1$, typically $d_2 = 2, 3, 4$ is very small and $d_1 = \frac{d_2}{n_1} n_2$, to obtain the complexity $O(n_2^2)$.

### 4.2 XoRA: EXPANDER LOW-RANK ADAPTATION

In the proposed method XoRA, structural sparsity is achieved by introducing sparse expander masked low-rank matrices $\tilde{A}, \tilde{B}$, where only the non-masked weights in these matrices are trainable. During backpropagation, only these weights receive gradient updates.

Using the methodology described in described in Section 4.1, we generate two bipartite expander graphs $G_A(V_{A_1}, V_{A_2}, E_A)$ and $G_B(V_{B_1}, V_{B_2}, E_B)$. For the graphs $G_A$ and $G_B$ we have the following cardinality properties: $G_A : |V_{A_1}| = r$, $|V_{A_2}| = k$, $G_B : |V_{B_1}| = d$, $|V_{B_2}| = r$, $E_A \subseteq V_{A_1} \times V_{A_2}$ and $E_B \subseteq V_{B_1} \times V_{B_2}$. We also ensure that $n_1 \times d_1 = n_2 \times d_2$. Where $n_1$ and $n_2$ are the cardinalities of the two vertex sets $V_A, V_B$, and $d_1$ and $d_2$ are their respective degrees.

Two expander masks $M_A \in \{0, 1\}^{r \times k}$ for matrix $A$, and $M_B \in \{0, 1\}^{d \times r}$ for matrix $B$ are used for adaptation. The expander masks are defined using the expander graphs $G_A(V_{A_1}, V_{A_2}, E_A)$ and $G_B(V_{B_1}, V_{B_2}, E_B)$ as:

$$M_{A_{ij}} = \begin{cases} 1 & \text{if } (i, j) \in E_A \\ 0 & \text{otherwise} \end{cases}, \quad M_{B_{ij}} = \begin{cases} 1 & \text{if } (i, j) \in E_B \\ 0 & \text{otherwise} \end{cases} \tag{2}$$

Sparse trainable low-rank matrices are created by applying the expander masks to the original low-rank matrices: $\tilde{B} = M_B \odot B$, $\tilde{A} = M_A \odot A$, where $\odot$ denotes the Hadamard (element-wise) product. The forward pass use the sparse expander masked trainable matrices:

$$h = Wx + \tilde{B}\tilde{A}x \tag{3}$$

Computed gradients are applied only for the trainable elements as determined by the expander masks:

$$\nabla_{A_{ij}} \mathcal{L}(\theta) = \begin{cases} \nabla_{\tilde{A}_{ij}} \mathcal{L}(\theta) & \text{if } M_{A_{ij}} = 1 \\ 0 & \text{if } M_{A_{ij}} = 0 \end{cases}, \quad \nabla_{B_{ij}} \mathcal{L}(\theta) = \begin{cases} \nabla_{\tilde{B}_{ij}} \mathcal{L}(\theta) & \text{if } M_{B_{ij}} = 1 \\ 0 & \text{if } M_{B_{ij}} = 0 \end{cases} \tag{4}$$

The objective function in XoRA is similar to the original loss function $\mathcal{L}(\theta)$ ($\theta$ represents the base model parameters), but here update is constrained to the sparse expander masked weights as follows.

$$A_{ij} \leftarrow \begin{cases} A_{ij} - \eta \nabla_{\tilde{A}_{ij}} \mathcal{L}(\theta) & \text{if } M_{A_{ij}} = 1 \\ A_{ij} & \text{if } M_{A_{ij}} = 0 \end{cases}, \quad B_{ij} \leftarrow \begin{cases} B_{ij} - \eta \nabla_{\tilde{B}_{ij}} \mathcal{L}(\theta) & \text{if } M_{B_{ij}} = 1 \\ B_{ij} & \text{if } M_{B_{ij}} = 0, \end{cases} \tag{5}$$

where $\eta$ is the learning rate and $\mathcal{L}(\theta)$ is the loss function. The structural sparsity of expander masks helps XoRA to significantly reduce the number of trainable parameters and improve generalization.

## 5 EXPERIMENTAL RESULTS

### 5.1 DATASETS AND EXPERIMENTAL SETUP

Evaluation of the proposed XoRA method is done on the General Language Understanding Evaluation (GLUE) benchmark Wang (2018) using RoBERTa base and RoBERTa large models (Liu, 2019). Due to computational limitations we did limited number of experiments on the resource and time intensive tasks MNLI, QQP and QNLI. The following GLUE benchmark tasks and evaluation metrics are reported in our study. **CoLA** (Corpus of Linguistic Acceptability): Matthews Correlation Coefficient, **SST-2** (Stanford Sentiment Treebank): Accuracy, **MRPC** (Microsoft Research Paraphrase Corpus): Accuracy, **STS-B** (Semantic Textual Similarity Benchmark): Pearson correlation, **RTE** (Recognizing Textual Entailment): Accuracy. For each a higher value is better.

Since we do not fine-tune MNLI, the MNLI initialization trick which involves fine-tuning the model on the MNLI dataset before fine-tuning on the target task (MRPC , STSB and RTE) is also not used. For RoBERTa base model, experiments are reported for MRPC, STS-B, and RTE with LoRA without the MNLI trick (LoRA•) for a fairer comparison with XoRA. Without the MNLI trick, the performance difference for MRPC and STS-B is less pronounced. However RTE suffers more without the MNLI trick, likely due to the small training set. For RoBERTa-large, the original LoRA paper reported metrics both with and without the MNLI trick (LoRA° and LoRA•). We used RoBERTa-base and RoBERTa-large from Hugging Face with the same setup as in the original LoRA paper for our initial experiments. Sparsification is performed only for the LoRA matrices corresponding to the Query (Q) and Value (V) layers. We perform 5 runs with different random seeds, recording the best epoch's outcome for each run. The median and standard deviation of these values are reported. The same hyperparameters as in the original LoRA paper (Hu et al., 2021) is used as shown in Table 12.

The expander mask configurations used in our experiments are shown in Table 11. Here, sparsity is defined as the ratio of number of zero elements in the masked LoRA matrices to the total number of elements. Note that, the sparsity levels can be varied as we consider LoRA matrices with different ranks. Maximum sparsity levels achieved by the expander mask generation process for a particular rank configuration is mentioned in Table 10. XoRA variant with the highest sparsity (75%) is used for fine-tuning in our experiments for the RoBERTa base and RoBERTa-large (rank-8). In the case of a rank-8 configuration, this 75% sparsity is the maximum achievable structured sparsity from an expander. For higher ranks, such as rank 32, the maximum structured sparsity from the expander mask would be higher (93.75%).

## 5.2 RESULTS AND DISCUSSION

### 5.2.1 COMPARISON BETWEEN RANDOM MASKING AND EXPANDER MASKING

Table 1 compares Randomly masked LoRA and XoRA performance on MRPC (Accuracy) and RTE(Accuracy) tasks for RoBERTa base model. XoRA is shown at different sparsity levels: 50%, 62.5%, and 75%. It is observed that the expander masks outperform the random masks at a high sparsity level. The random masking method has a high variability of performance for the 5 runs, whereas the expander mask provides a stable performance over these runs. Especially at higher sparsity levels the random masked LoRA is unstable and performance drop sharply. XoRA has consistent and stable performance across all sparsity levels. The XoRA variant with 75% sparsity is selected for further experiments due to its efficient parameter usage ($0.075M$ trainable parameters) while maintaining performance close to LoRA. Some key observations from the results are:

- At 50% sparsity ($0.15M$ parameters), it outperforms LoRA's MRPC accuracy ($89.7\pm0.6$) and matches RTE accuracy ($78.7\pm0.9$).
- At 62.5% sparsity ($0.1125M$ parameters), it has competitive performance with LoRA.
- Even at 75% sparsity ($0.075M$ parameters), it maintains performance close to LoRA on MRPC ($89.5 \pm 0.7$) and RTE ($76.9 \pm 1.3$).
- At all the sparsity levels XoRA outperforms the randomly masked LoRA. Also it has lower variability than random masking.

Table 1: Comparison of randomly masked LoRA and XoRA for MRPC and RTE tasks using the RoBERTa base model.

| Method | Trainable Parameters | Sparsity Level | MRPC (Acc) | RTE (Acc) |
|---|---|---|---|---|
| FT | 125M | - | 90.2 | 91.2 |
| LoRA• | 0.3M | 0% | 89.5±0.8 | 78.7±1.3 |
| Random | 0.15M | 50% | 87.3±2.5 | 75.5±2.8 |
| Random | 0.075M | 75% | 85.3±3.4 | 73.3±2.2 |
| XoRA | 0.15M | 50% | 89.7±0.6 | 78.7±0.9 |
| XoRA | 0.1125M | 62.5% | 89.2±0.9 | 77.6±1.3 |
| XoRA | 0.075M | 75% | 89.5±0.7 | 76.9±1.3 |

### 5.2.2 COMPARISON BETWEEN XORA AND OTHER ADAPTATION METHODS

We now compare the performance of XoRA with LoRA and other parameter-efficient fine-tuning (PEFT) baselines for the RoBERTa models on the GLUE tasks. The methods compared are FT (Full fine-tuning), BitFit (Zaken et al., 2021), Adpt$^D$ (Rücklé et al., 2020), Adpt$^H$ (Houlsby et al., 2019), Adpt$^P$ (Pfeiffer et al., 2020), LoRA-FA (Zhang et al., 2023a), and LoRA (Hu et al., 2021).

Tables 2 and 3 presents GLUE benchmark results for the RoBERTa base and RoBERTa large models respectively. Results of all methods except XoRA are sourced from prior work (Hu et al. (2021); Zhang et al. (2023b)). For RoBERTa base model, we repeated the LoRA experiments for MRPC, STS-B, and RTE without the MNLI trick (LoRA•) for a fairer comparison with XoRA.

Table 2: Performance comparison of XoRA and other adaptation methods on the GLUE benchmark for RoBERTa base.

| Method | Trainable Parameters | SST-2 (Acc) | CoLA (MCC) | MRPC (Acc) | STS-B (Pear) | RTE (Acc) | Avg. |
|---|---|---|---|---|---|---|---|
| FT | 125M | 94.8 | 63.6 | 90.2 | 91.2 | 78.7 | 83.7 |
| BitFit | 0.1M | 93.7 | 62.0 | 92.7 | 90.8 | 81.5 | 84.1 |
| Adpt$^D$ | 0.3M | 94.2±0.1 | 60.8±0.4 | 88.5±1.1 | 89.7±0.3 | 71.5±2.7 | 80.9 |
| Adpt$^D$ | 0.9M | 94.7±0.3 | 62.6±0.9 | 88.4±0.1 | 90.3±0.1 | 75.9±2.2 | 82.4 |
| LoRA° | 0.3M | 95.1±0.2 | 63.4±1.2 | 89.7±0.7 | 91.5±0.2 | 86.6±0.7 | 85.3 |
| LoRA• | 0.3M | 95.1±0.2 | 63.4±1.2 | 89.5±0.8 | 90.1±0.2 | 78.7±1.3 | 83.4 |
| VeRA* | 0.043M | 94.6±0.1 | 65.6±0.8 | 89.5±0.5 | 90.7±0.2 | 78.7±0.7 | 83.8 |
| **XoRA** | **0.075M** | 94.8±0.2 | 62.5±0.9 | 89.5±0.7 | 90.1±0.3 | 76.9±1.3 | 82.8 |

Table 3: Performance comparison of XoRA and other adaptation methods on the GLUE benchmark for RoBERTa large. *For VeRA, we report the scores published in (Kopiczko et al., 2023). Our replication experiments yielded slightly lower averages: 83.0 for RoBERTa-base and 86.2 for RoBERTa-large (due to seeds and variation in hyperparameters)

| Method | Trainable Parameters | SST-2 (Acc) | CoLA (MCC) | MRPC (Acc) | STS-B (Pear) | RTE (Acc) | Avg. |
|---|---|---|---|---|---|---|---|
| FT | 355.0M | 96.4 | 68.0 | 90.9 | 92.4 | 86.6 | 86.9 |
| Adpt$^P$ | 3.0M | 96.1±0.3 | 68.3±1.0 | 90.2±0.7 | 92.1±0.7 | 83.8±2.9 | 86.1 |
| Adpt$^P$ | 0.8M | 96.6±0.2 | 67.8±2.5 | 89.7±1.2 | 91.9±0.4 | 80.1±2.9 | 85.2 |
| Adpt$^H$ | 6.0M | 96.2±0.3 | 66.5±4.4 | 88.7±2.9 | 91.0±1.7 | 83.4±1.1 | 85.2 |
| Adpt$^H$ | 0.8M | 96.3±0.5 | 66.3±2.0 | 87.7±1.7 | 91.5±0.5 | 72.9±0.5 | 82.9 |
| LoRA-FA | 3.7M | 96.0 | 68.0 | 90.0 | 92.0 | 86.1 | 86.4 |
| LoRA° | 0.8M | 96.2±0.5 | 68.2±1.9 | 90.9±1.2 | 92.6±0.2 | 87.4±2.5 | 87.1 |
| LoRA• | 0.8M | 96.2±0.5 | 68.2±1.9 | 90.2±1.0 | 92.3±0.5 | 85.2±1.1 | 86.4 |
| VeRA* | 0.061M | 96.1±0.1 | 68.0±0.8 | 90.9±0.7 | 91.7±0.8 | 85.9±0.7 | 86.5 |
| **XoRA** | **0.2M** | 96.1±0.1 | 67.8±1.6 | 90.0±0.6 | 91.9±0.2 | 85.6±1.3 | 86.3 |

Using only about 25% of the trainable parameters of LoRA, the proposed method attains comparable performance across GLUE tasks. At a very high sparsity XoRA's average score 82.8 and 86.3, is only 0.6 and 0.1 lower than LoRA for RoBERTa base and RoBERTa large respectively. This underscores the effectiveness of using structured sparsity from expander graphs. The proposed method is comparable to the state of the art at high sparsity.

### 5.2.3 INSTRUCTION TUNING

We performed instruction tuning on the LLaMA 2 7B base model (Touvron et al., 2023) using the Alpaca dataset (Taori et al., 2023). The original Alpaca dataset comprises of $52k$ instruction-response pairs generated from OpenAI's text-davinci-003 model following self-instruction techniques introduced by (Wang et al., 2022), We specifically use the GPT-4 variant of it (Alpaca-GPT4[1]). We

---
[1]https://huggingface.co/datasets/vicgalle/alpaca-gpt4

perform fine-tuning using both XoRA and LoRA. The tuning was applied across all attention layers (Query, Key, Value and Projection layers). Detailed hyperparameters are summarized in Table 13.

We evaluate our instruction-finetuned XoRA and LoRA models using EleutherAI's lm-eval-harness Gao et al. (2024) to quantitatively assess their factual accuracy and knowledge retention. While instruction fine-tuning improves instruction-following capabilities, models often overfit to instruction patterns at the cost of general knowledge and factual accuracy. To assess this trade-off, we use TruthfulQA to measure factual correctness and MMLU global facts for general knowledge retention.

XoRA demonstrates consistent improvements over LoRA across all metrics - achieving up to **2.3%** gain on TruthfulQA-MC1, **3%** gain on TruthfulQA-MC2, and **3%** on MMLU global facts while using only **2.1M** parameters ($\downarrow$ **96.9%** parameter reduction at rank $r = 64$). The performance gains can be attributed to XoRA's implicit regularization during fine-tuning, helping preserve model's factual knowledge while adapting to instruction following. This is particularly evident in Truth-fulQA results where XoRA models show consistent improvements across increasing ranks, suggesting better retention of factual knowledge compared to LoRA variants. The improved performance on MMLU global facts further supports that XoRA helps maintain domain-specific factual knowledge while adapting to instruction following, effectively addressing the common trade-off between instruction coherence and factual accuracy in instruction fine-tuned models.

Table 4: Comparison of Alpaca Fine-tuned LoRA and XoRA on Llama 2 7B base model across different ranks $(r)$, and performance on TruthfulQA (TQA) and MMLU Global Fact benchmarks.

| Model | Params | TQA-MC1 | TQA-MC2 | MMLU |
|---|---|---|---|---|
| Llama 2 7B base | 6.7B | 25.3% | 39.7% | 32.0% |
| LoRA $r$=8 | 8.4M | 29.1% | 43.6% | 26.0% |
| XoRA $r$=8 | 2.1M ($\downarrow$ 75.0%) | **30.4%** | **44.5%** | **28.0%** |
| LoRA $r$=16 | 16.8M | 30.8% | 46.0% | 27.0% |
| XoRA $r$=16 | 2.1M ($\downarrow$ 87.5%) | **31.6%** | **46.6%** | **29.0%** |
| LoRA $r$=32 | 33.6M | 30.2% | 44.1% | 30.0% |
| XoRA $r$=32 | 2.1M ($\downarrow$ 93.8%) | **32.1%** | **47.3%** | **30.0%** |
| LoRA $r$=64 | 67.1M | 30.5% | 45.4% | 29.0% |
| XoRA $r$=64 | 2.1M ($\downarrow$ 96.9%) | **32.8%** | **48.4%** | **32.0%** |

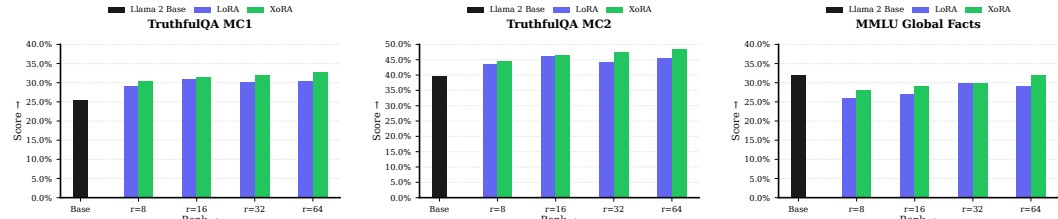

Figure 2: Performance comparison of Alpaca Fine-tuned Llama 2 7B Base LoRA and XoRA across different ranks (r) on: (left) TruthfulQA MC1, (middle) TruthfulQA MC2, and (right) MMLU Global Facts benchmarks.

We further validate XoRA's effectiveness through MT-Bench (Zheng et al., 2023) evaluation using GPT-4 (OpenAI, 2023) as judge, which confirms the quantitative improvements translate to better qualitative performance.

Table 5: Comparison on MT-Bench for Alpaca fine-tuned Llama 2 7B using LoRA and XoRA.

| Method | Number of Parameters | MT-Bench Score |
|---|---|---|
| LoRA (r=64) | 67.1M | 5.4 |
| **XoRA (r=64)** | **2.1M** | **5.7** |

In instruction fine-tuning evaluations, VeRA exhibits lower performance compared to LoRA across various model scales as reported in the paper (Kopiczko et al., 2023), see Table 6. In contrast, XoRA

demonstrates superior instruction-following capabilities compared to its LoRA counterpart despite using significantly fewer parameters as shown in Table 5.

Table 6: Comparison of MT-Bench results for Alpaca fine-tuned Llama 2 7B using LoRA and VeRA.

| Method | Number of Parameters | MT-Bench Score |
|--------|---------------------|----------------|
| LoRA | 159.9M | 5.19 |
| VeRA | **1.6M** | 5.08 |

### 5.2.4 IMPACT OF XoRA ON MEMORY USAGE AND TRAINING TIME

XoRA introduces expander masked low ranks, meaning that many of the low-rank parameters are masked out and do not receive gradients during fine-tuning. This significantly reduces the number of parameters that need to be stored and updated during training. For a rank $64$ configuration with XoRA we can achieve $96.9\%$ parameter reductions. To assess the training time and GPU memory advantages of our method, we compared LoRA and XoRA while instruction fine-tuning the LLaMA 2 7B model on the Alpaca dataset with the same rank of $64$. Our results 7 show a $9.2\%$ reduction in GPU memory usage with XoRA compared to LoRA, as well as a $3.4\%$ reduction in training time. In comparison VeRA (Kopiczko et al., 2023) includes more operations than LoRA due to the additional vector multiplies in the forward pass, and results in a $1.8\%$ increase in training time and a $7.4\%$ reduction GPU memory.

Table 7: LoRA vs XoRA Memory Usage And Training Time Comparison at Rank 64

| **Metric** | **LoRA** | **XoRA** | **Reduction** |
|------------|----------|----------|---------------|
| Peak Memory | 18.65 GB | **16.94 GB** | **9.2%** |
| Training Time | 10522 s | **10169 s** | **3.4%** |

In XoRA, because many of the parameters are masked and do not participate in the gradient computation, the GPU memory requirement for storing gradients is reduced.

### 5.2.5 ADDITIONAL EXPERIMENTS

Table 8: Comparison of XoRA with existing parameter-efficient fine-tuning methods on GLUE benchmark tasks using DeBERTa-V3-base as the backbone model. Results of all the baseline methods are taken from prior work (Ding et al., 2023a). Results of XoRA on DeBERTA-V3-base are obtained using the same hyperparams and experimentation setups described in (Ding et al., 2023a). Average test results along with standard deviation over 5 random seed is reported.

| **Method** | **#Params** | **MRPC** | **STS-B** | **RTE** | **CoLA** | **SST-2** | **Avg.** |
|------------|-------------|----------|-----------|---------|----------|-----------|----------|
| Fine-Tune | 184M | $89.22_{\pm 0.69}$ | $91.59_{\pm 0.47}$ | $82.49_{\pm 1.48}$ | $69.21_{\pm 2.24}$ | $95.64_{\pm 0.52}$ | 85.63 |
| Adapter | 1.41M | $89.90_{\pm 2.10}$ | $92.21_{\pm 0.33}$ | $82.44_{\pm 1.74}$ | $69.00_{\pm 0.91}$ | $95.16_{\pm 0.46}$ | 85.74 |
| Bitfit | 0.1M | $87.16_{\pm 0.58}$ | $89.71_{\pm 0.58}$ | $76.12_{\pm 1.54}$ | $68.70_{\pm 1.85}$ | $94.38_{\pm 0.28}$ | 83.21 |
| LoRA | 1.33M | $89.71_{\pm 1.32}$ | $91.86_{\pm 0.29}$ | $85.32_{\pm 0.86}$ | $69.73_{\pm 1.42}$ | $95.57_{\pm 0.21}$ | 86.44 |
| AdaLoRA | 1.27M | $90.22_{\pm 0.40}$ | $91.39_{\pm 0.25}$ | $87.36_{\pm 0.30}$ | $70.86_{\pm 1.43}$ | $\mathbf{95.95}_{\pm 0.37}$ | 87.16 |
| SoRA | 0.91M | $91.98_{\pm 1.16}$ | $\mathbf{92.22}_{\pm 0.24}$ | $87.77_{\pm 1.56}$ | $71.48_{\pm 1.17}$ | $95.64_{\pm 0.23}$ | 87.82 |
| **XoRA** | **0.33M** | $\mathbf{92.02}_{\pm 1.22}$ | $92.21_{\pm 0.18}$ | $\mathbf{87.91}_{\pm 1.07}$ | $\mathbf{71.99}_{\pm 1.25}$ | $95.92_{\pm 0.29}$ | **88.01** |

To evaluate XoRA's effectiveness on natural language generation tasks (NLG), we conducted experiments on the E2E NLG Challenge benchmark (Novikova et al., 2017) using GPT-2 Medium as the base model. We report the best epochs outcome as in the LoRA paper (Hu et al., 2021). **XoRA demonstrates strong empirical performance while utilizing only half of LoRA's parameter count**, maintaining competitive scores across BLEU (70.4), NIST (8.86), and ROUGE-L (71.7).

Table 9: Comparison of different adaptations methods for GPT-2 Medium on the E2E NLG Challenge benchmark. Results marked with [†] are from the original LoRA paper (Hu et al., 2021), [‡] from the DyLoRA paper (Valipour et al., 2022), and [⋆] taken from the VeRA paper (Kopiczko et al., 2023).

| Method | #Params | BLEU | NIST | METEOR | ROUGE-L | CIDEr |
|---|---|---|---|---|---|---|
| GPT2-M FT[†] | 354.92M | 68.2 | 8.62 | 46.2 | 71.0 | 2.47 |
| AdptL[†] | 0.37M | 66.3 | 8.41 | 45.0 | 69.8 | 2.40 |
| AdptL[†] | 11.09M | 68.9 | 8.71 | 46.1 | 71.3 | 2.47 |
| AdptH[†] | 11.09M | 67.3 | 8.50 | 46.0 | 70.7 | 2.44 |
| FTTop2[†] | 25.19M | 68.1 | 8.59 | 46.0 | 70.8 | 2.41 |
| PreLayer[†] | 0.35M | 69.7 | 8.81 | 46.1 | 71.4 | 2.49 |
| LoRA[†] | 0.35M | **70.4** | 8.85 | **46.8** | **71.8** | **2.53** |
| DyLoRA[‡] | 0.39M | 69.2 | 8.75 | 46.3 | 70.8 | 2.46 |
| AdaLoRA[⋆] | 0.38M | 68.2 | 8.58 | 44.1 | 70.7 | 2.35 |
| VeRA[⋆] | 0.098M | 70.1 | 8.81 | 46.6 | 71.5 | 2.50 |
| **XoRA** | 0.196M | **70.4** | **8.86** | 46.4 | 71.7 | 2.51 |

## 6 LIMITATIONS AND DISCUSSIONS

One of the limitations is that, Ramanujan graphs are (rigorously) defined in the regular or biregular case, with degree $\geqslant 2$. The degree consideration forces the sparse mask based on it to concentrate only in sparsification of edges and not that of vertices (vertex dropout). Further, the regularity forces a lower bound on the number of parameters. Thus, the method sparsifies the most when the LoRA rank is large. For instance when $r = 64$ we can achieve 97% sparsity (as given in table 7). Another point is that random generation theoretically always guarantees an expander mask but not a Ramanujan mask. This is because random biregular bipartite graphs are known to be expanders but not Ramanujan (the fact that they are ramanujan is a conjecture). That said practically we do get Ramanujan graphs because it has been shown that will be weakly Ramanujan by Freidman, in his proof of Alon's conjecture see (Friedman, 2003). Note that for XoRA, whenever we generate random expanders, we do an additional check if it is Ramanujan or not and then proceed. The existence of biregular Ramanujan graphs of all degrees have been shown by the works of Marcus–Spielman–Srivastava (Marcus et al., 2015; 2018). For the deterministic generation it is not a problem. Currently we apply the technique of Lubotzky–Phillips–Sarnak (Lubotzky et al., 1988), and Hall–Pruder–Sawin (Hall et al., 2018) (Ramanujan $r$-coverings). Another deterministic construction can be achieved using the technique of Gribinski–Marcus (Gribinski & Marcus, 2021).

## 7 CONCLUSION

In this work, we introduce XoRA (Expander-based Low-Rank Adaptation), a novel approach that integrates structural sparsity into the low-rank matrices of the LoRA adaptation method using bipartite expander graphs. XoRA effectively addresses the over-parameterization often present in low-rank update matrices, by masking majority of the elements.

The proposed XoRA method achieves comparable or superior performance to LoRA and other state of the art sparsification techniques (as can be seen from Tables 1, 2, 3, 4, 5, 8 and 9) while utilizing significantly fewer parameters. This efficiency is particularly valuable in resource-constrained computational environments. Our experiments show that XoRA exhibits robust performance at higher sparsity levels compared to random masking. The expander graph structure ensures maintained connectivity of the network despite a high sparsity and thus preserving the performance.

The expander masking inherent in XoRA offers regularization benefits during the fine-tuning process. This can improve generalization and reduce overfitting. The XoRA approach shows promise for integration with other parameter-efficient fine-tuning techniques, potentially leading to even greater parameter efficiency and adaptability.

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

## A    APPENDIX: EXPANDER GRAPHS AND RAMANUJAN GRAPHS

Expander graphs are highly connected, sparse graphs defined using the notion of Cheeger constants. The vertex Cheeger constant has been previously defined.

Cheeger constants can also be defined using the notion of edges or spectrum of adjacency matrices giving rise to edge expanders, and spectral expanders. When we consider the *edge boundary* of a set of vertices, which is defined as the set of edges with one endpoint in a vertex subset $X \subset V(\Gamma)$ and the other endpoint outside of $X$, it leads us to the definition of the **edge Cheeger constant** $\mathbf{h}_E(\Gamma)$. See Figure 3 for a graphical representation of the best cut [2] in a 12 by 8 bipartite graph which achieves the edge Cheeger constant.

These two constants, which measure edge expansion and vertex expansion, are related by the following inequality:

$$\frac{\mathbf{h}_V(\Gamma)}{D} \leq \mathbf{h}_E(\Gamma) \leq \mathbf{h}_V(\Gamma),$$

where $D$ represents the maximum degree of the graph. This relationship demonstrates that the notions of vertex expansion and edge expansion are closely related, allowing us to use them interchangeably in many contexts.

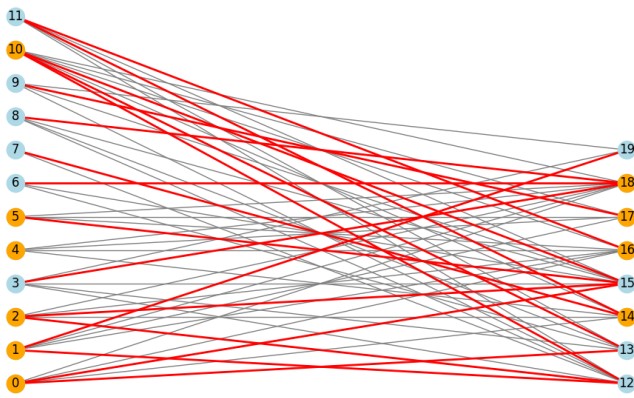

Figure 3: Example of a "cut" achieving the edge Cheeger constant in a 12 by 8 bipartite graph. The subset $S$ of cardinality 10 is pointed out in orange. The red edges have one vertex is $S$ and the other in $V \setminus S$. The edge Cheeger constant is the number of red edges divided by $D|S|$.

In the literature, having a high Cheeger constant is often referred to as possessing high *combinatorial expansion*. Intuitively, a graph with a large vertex (or edge) Cheeger constant is difficult to "separate," and is a desirable quality for free flow of information. However, it is important to note that computing the Cheeger constants of graphs is generally an NP-hard problem, which makes it impractical to directly calculate in most real-world applications.

To overcome the difficulty of computing Cheeger constants, *spectral methods* are often employed. These techniques rely on the eigenvalues of the graph's adjacency matrix, which contain significant information about the graph's structural properties, including its expansion capabilities. Given a finite undirected graph $\Gamma$, let the eigenvalues of its adjacency matrix be ordered as $\lambda_n \leq \lambda_{n-1} \leq \cdots \leq \lambda_1$. These eigenvalues are real, and for any graph, $\lambda_1 \leq D$, where $D$ is the maximum degree of the graph. Importantly, equality holds, $\lambda_1 = D$, if and only if the graph is D-regular, meaning every vertex in the graph has exactly $D$ edges incident to it.

For a graph to be D-regular implies that all vertices have the same degree, which contributes to its regular structure. In the special case of *bipartite graphs*, the graph is said to be D-regular if it consists of two sets of vertices, where every vertex in each set has exactly $D$ edges connected to vertices in the other set, maintaining equal degrees across both partitions.

---

[2]In graph theory, a cut is a partition of the vertices of a graph into two disjoint subsets. Any cut determines a cut-set, the set of edges that have one endpoint in each subset of the partition.

Graphs are said to be *spectral expanders* if their spectral properties imply strong connectivity and expansion. Specifically, a graph $\Gamma$ is considered a spectral expander if the quantities $|\lambda_1 - \lambda_2|$ and $|\lambda_1 - \lambda_k|$ are bounded away from zero, where $k = n - 1$ for bipartite graphs and $k = n$ otherwise. These bounds on the spectral gap ensure that the graph exhibits good expansion properties, as the eigenvalues provide insight into the graph's ability to resist being "cut" into disconnected components.

**Question A.1.** *Why Is Strong Spectral Expansion Important?*

The significance of spectral expansion lies in its connection to *combinatorial expansion*. The relationship between spectral and combinatorial expansion is formalized through the *discrete Cheeger-Buser inequality*, which establishes that spectral expansion implies combinatorial expansion.

Suppose $\Gamma$ is a $d$-regular graph. The largest eigenvalue is $\lambda_1 = d$. Then the *Cheeger constant* $\mathbf{h}_E(\Gamma)$ satisfies the following inequalities known as the discrete Cheeger–Buser inequalities:

$$\frac{1}{2}(d - \lambda_2) \leq \mathbf{h}_E(\Gamma) \leq \sqrt{2d(d - \lambda_2)}.$$

See Alon (1986); Hoory et al. (2006) for details. These inequalities establish a relationship between the spectral properties of the graph (through the second largest eigenvalue $\lambda_2$ of the adjacency matrix) and its combinatorial expansion properties (through the Cheeger constant $h(G)$). The lower bound is often referred to as Cheeger's inequality, while the upper bound is known as Buser's inequality in the discrete setting.

**Lower Bound (Cheeger's Inequality):**

$$h(G) \geq \frac{1}{2}(d - \lambda_2).$$

This inequality implies that if the second largest eigenvalue $\lambda_2$ is close to $d$, then the graph has poor expansion properties.

**Upper Bound (Buser's Inequality):**

$$h(G) \leq \sqrt{2d(d - \lambda_2)}.$$

This indicates that a significant gap between $\lambda_2$ and $d$ leads to strong expansion properties.

In practice, this means that the larger the spectral expansion, the more difficult it is to isolate subsets of vertices, which corresponds to greater connectivity and robustness of the graph structure.

Since computing combinatorial expansion directly is NP-hard, leveraging spectral expansion allows us to sidestep this difficulty. Spectral techniques offer an efficient way to measure and guarantee expansion properties. Naturally, a key question arises: *Is there a limit to how large the spectral expansion can become, or can it be arbitrarily large?*

This leads us to the concept of **Ramanujan graphs**, which are the *optimal spectral expanders*. Ramanujan graphs achieve the best possible trade-off between low degree and high expansion properties. In fact, these graphs are known to provide the largest possible spectral gap for a given degree, making them the ideal candidates for networks and systems that require both sparse connectivity and strong expansion. Figure 4 visualizes a typical bipartite Ramanujan graph.

It is essential to note that *disconnected graphs* are not considered expanders. The reason for this is straightforward: in a disconnected graph, the boundary of any connected component is empty, meaning there is no flow of information between different parts of the graph. The Cheeger constant, as an expansion parameter, effectively measures how well-connected the graph is, and thus a disconnected graph has zero expansion. In contrast, a graph with a high Cheeger constant, or equivalently, a large spectral gap, exhibits strong expansion, meaning that it remains well-connected even after the removal of some edges or vertices.

At the extreme, the **complete graph** represents the best possible expander, as it has the maximum possible connectivity. However, the complete graph also has the highest possible degree, which makes it impractical in many applications that require sparse connections. Therefore, a "good expander" is one that balances *low degree* with *high expansion* properties. Ramanujan graphs serve as a prime example of such an optimal balance, making them highly valuable in both theoretical and practical contexts where efficient and robust network structures are needed.

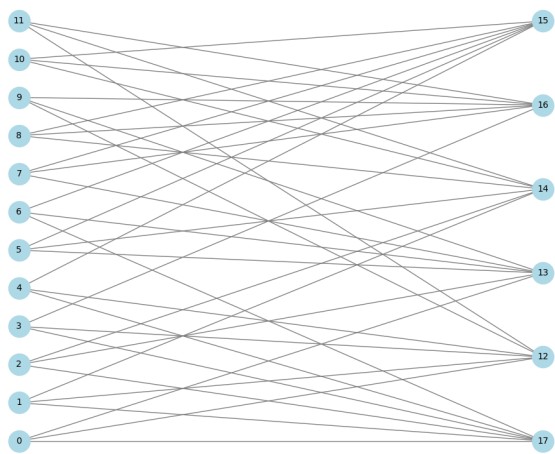

Figure 4: Example of a 12 by 6 bipartite (3,6) biregular Ramanujan graph.

## B   APPENDIX : HYPERPARAMETERS

Table 10: Maximum sparsity levels for bipartite expander graphs with varying ranks and expander sizes. The maximum sparsity is achieved (left degree $d_L = 2$) when number of edges are minimized while maintaining the expander properties.

| Layer Size | LoRA Rank | Expander Size $(d_L, d_R)$ | Max Sparsity | Trainable Param |
|---|---|---|---|---|
| $768 \times 768$ | 8 | $768 \times 8$ (2, 192) | 75% (6/8) | 25% (2/8) |
| $768 \times 768$ | 16 | $768 \times 16$ (2, 96) | 87.5% (14/16) | 12.5% (2/16) |
| $768 \times 768$ | 32 | $768 \times 32$ (2, 48) | 93.75% (30/32) | 6.25% (2/32) |
| $768 \times 768$ | 64 | $768 \times 64$ (2, 24) | 96.88% (62/64) | 3.12% (2/64) |
| $1024 \times 1024$ | 8 | $1024 \times 8$ (2, 256) | 75% (6/8) | 25% (2/8) |
| $1024 \times 1024$ | 16 | $1024 \times 16$ (2, 128) | 87.5% (14/16) | 12.5% (2/16) |
| $1024 \times 1024$ | 32 | $1024 \times 32$ (2, 64) | 93.75% (30/32) | 6.25% (2/32) |
| $1024 \times 1024$ | 64 | $1024 \times 64$ (2, 32) | 96.88% (62/64) | 3.12% (2/64) |

Table 11: Bipartite expander mask configuration for rank-8 low-rank matrices in LoRA. Matrices for the Query and Value layers are sparsified. Expander Size refers to the number of vertices of the corresponding bipartite expander graphs. The numbers $(d_L, d_R)$ indicates degrees of the $d_L$-left-regular and $d_R$-right-regular bipartite graphs.

| Model | Layer Size | Expander Size $(d_L, d_R)$ | Sparsity |
|---|---|---|---|
| RoBERTa Base | $768 \times 768$ | $768 \times 8$ (2, 192) | 75.0% |
| RoBERTa Base | $768 \times 768$ | $768 \times 8$ (3, 288) | 62.5% |
| RoBERTa Base | $768 \times 768$ | $768 \times 8$ (4, 384) | 50.0% |
| RoBERTa Large | $1024 \times 1024$ | $1024 \times 8$ (2, 256) | 75.0% |
| RoBERTa Large | $1024 \times 1024$ | $1024 \times 8$ (3, 384) | 62.5% |
| RoBERTa Large | $1024 \times 1024$ | $1024 \times 8$ (4, 512) | 50.0% |

The hyperparameters used in the alpaca fine-tuning of Llama 2 7B using LoRA and XoRA are discussed in 13.

Table 12: Hyperparameters for RoBERTa base XoRA / RoBERTa large XoRA, on GLUE benchmark.

| Task | Batch Size | Epochs | Learning Rate |
|------|-----------|--------|---------------|
| SST-2 | 16 / 4 | 60 / 10 | 5e-4 / 4e-4 |
| MRPC | 16 / 4 | 30 / 20 | 4e-4 / 3e-4 |
| STS-B | 16 / 4 | 40 / 10 | 4e-4 / 2e-4 |
| RTE | 32 / 4 | 80 / 20 | 5e-4 / 4e-4 |
| CoLA | 32 / 4 | 80 / 20 | 4e-4 / 2e-4 |

| | |
|---|---|
| Optimizer: | AdamW |
| Warmup ratio: | 0.06 |
| LR schedule: | Linear |
| Max sequence length: | 512 (RoBERTa base) / 128 (RoBERTa large) |
| LoRA config: | $r_q = r_v = 8$, $\alpha = 8$ (base) / 4 (large) |

Table 13: Hyperparameters for Alpaca Fine-tuning of Llama 2 7B using LoRA and XoRA

| Optimization Parameters | |
|---|---|
| Optimizer | AdamW |
| Learning Rate | 3e-4 |
| Weight Decay | 0.01 |
| **Training Schedule** | |
| Scheduler Type | Cosine |
| Total Training Examples | 52,000 (Alpaca dataset) |
| Training Iterations | 50,000 |
| Training Epoch | 1 |
| Warmup Steps | 100 |
| **Batch Configuration** | |
| Global Batch Size | 128 |
| Micro-batch Size | 1 |
| Gradient Accumulation Steps | 128 |
| **LoRA and XoRA Configurations** | |
| Ranks ($r$) | 8, 16, 32, 64 |
| Alpha Values ($\alpha = 2r$) | 16, 32, 64, 128 |
| LoRA Dropout | 0.05 |
| **Model Configuration** | |
| Base Model | Llama 2 7B |
| Adapted Layers | All attention layers (Q, K, V, and O) |
| Quantization Method | bnb.nf4 (4-bit Normal Float) |
| **Expander Mask Configuration** | |
| Expander Mask Dimensions | $4096 \times r$ and $r \times 4096$ |
| Left Degree | 2 |
| Sparsity | $(r - 2)/r$ |

The hyperparams and experimental setups for XoRA fine-tuning of DeBERTa-V3-base on glue benchmark are outlined here. We conduct experiments using DeBERTa-V3-base (He et al., 2021) from HuggingFace Transformers library. The hyperparameters for all experiments are detailed in Table 14. We report average test results with standard deviations across 5 random seeds. We kept the same experimental setup as in SoRA (Ding et al., 2023a) baselines, we handle dataset splits differently based on their sizes as in (Ding et al., 2023a). For smaller datasets (MRPC, RTE, CoLA, and STS-B; $n_{samples} < 10K$), we split the validation set in half to create test and validation sets. For the larger dataset (SST-2; $n_{samples} > 10K$), we reserve $1K$ samples from the training set for validation while using the original validation set as the test set. Performance is evaluated using task-specific metrics: F1 score for MRPC, Matthews Correlation Coefficient (MCC) for CoLA, Accuracy for SST-2 and RTE, and Pearson correlation for STS-B.

The hyperparams used for XoRA E2E benchmarks experiments on GPT2-M is detailed below in Table 15

## C  APPENDIX : INSTRUCTION TUNING EXPERIMENTS

In this section we outline the datasets and evaluation metrics used to assess XoRA's performance in instruction tuning scenarios as shown in 5.2.3 section presented previously.

We specifically focus on the EleutherAI's lm-eval-harness and MT-Bench framework and conduct our experiments using the Alpaca instruction tuning setup, which provides a standardized environment for comparing instruction-tuned models, including parameter-efficient methods.

TruthfulQA consist of diverse MCQ and open ended questions designed to test a model's knowledge and resistance to generating false hoods. It includes two main evaluation modes:

- MC1 (Single-true): In this setup, each question is paired with 4-5 answer choices, with only one correct answer. The model selects an answer by assigning the highest log-probability

Table 14: Hyperparameters for XoRA fine-tuning experiments on GLUE tasks using DeBERTa-V3-base.

| General Configuration | |
|---|---|
| Backbone Model | DeBERTa-V3-base |
| Max Sequence Length | 256 |
| Learning Rate Schedule | Linear |
| Warmup Ratio | 0.06 |
| Max Gradient Norm | 0.1 |
| **XoRA Configuration** | |
| LoRA Rank ($r$) | 16 |
| LoRA Alpha ($\alpha$) | 32 |
| **Expander Mask Configurations** | |
| Attention Layers | $768 \times 16$ and $16 \times 768$ |
| Intermediate Layer | $3072 \times 16$ and $16 \times 768$ |
| Output Layer | $768 \times 16$ and $16 \times 3072$ |
| Left Degree | 2 |
| Sparsity | 87.5% (14/16) |
| **Task-Specific Hyperparameters** | |
| Task (Metric) | {Learning Rate, Batch Size, Epochs} |
| CoLA (MCC) | {3e-4, 8, 20} |
| MRPC (F1) | {3e-4, 8, 20} |
| STS-B (Pearson) | {3e-4, 8, 20} |
| SST-2 (Accuracy) | {3e-4, 8, 10} |
| RTE (Accuracy) | {2e-3, 32, 50} |

of completion to one of the choices. Accuracy is measured as the percentage of questions for which the model selects the correct answer.

- MC2 (Multi-true): Here, each question is associated with a set of true and false reference answers. The model assigns probabilities to these answers, and the score is computed as the normalized total probability assigned to the set of true answers.

TruthfulQA emphasizes challenges in fact-based reasoning and penalizes models for hallucinating or overconfidently generating incorrect answers.

MT-Bench is a benchmark designed to evaluate large language models on their instruction-following abilities across 80 tasks covering a wide range of domains, such as coding, math, reasoning, factual knowledge, creative writing, and open-ended dialogue. Each task consists of two turns:

- (1) an initial instruction or query that the model must address
- (2) a follow-up question designed to further probe the model's understanding, reasoning, or ability to revise its initial response.

| E2E NLG Challenge Dataset | |
|---|---|
| Training Examples | 42,061 |
| Development Examples | 4,672 |
| Test Examples | 4,693 |
| Number of Attributes | 8 |
| Avg. MR Length | 5.43 |
| Avg. Reference Length | 22.1 |
| **GPT2-Medium Architecture** | |
| Total Parameters | 345M |
| Number of Layers | 24 |
| Hidden Size | 1024 |
| Attention Heads | 16 |
| Head Dimension | 64 |
| Feedforward Dimension | 4096 |
| Vocabulary Size | 50,257 |
| **XoRA Configuration** | |
| Rank (r) | 4 |
| Sparsity | 50% (2/r) |
| Adaptation | $r_q = r_v = 4$ |
| **Training Configuration** | |
| Optimizer | AdamW |
| Learning Rate | 2e-4 |
| Learning Rate Schedule | Linear |
| Weight Decay | 0.01 |
| Batch Size | 8 |
| Number of Epochs | 5 |
| Warmup Steps | 500 |
| Label Smoothing | 0.1 |
| LoRA Dropout | 0.1 |
| **Inference Configuration** | |
| Beam Size | 10 |
| Length Penalty | 0.9 |
| No Repeat N-gram Size | 4 |
| Max Length | 128 |
| Temperature | 1.0 |

Table 15: Hyperparameter configurations for XoRA experiments on the E2E NLG Challenge benchmark using GPT-2 Medium as the base model.

