# OpenReview forum: "XoRA: Expander adapted LoRA finetuning"
_ICLR.cc/2025/Conference — Submitted to ICLR 2025_

### Official Review · Reviewer_yii8 · 2024-10-29

**Soundness:** 2
**Presentation:** 2
**Contribution:** 2
**Rating:** 3
**Confidence:** 3

**Summary:**

The paper proposes an approach to sparsify LoRA's low-rank matrices to reduce redundancy. They induce sparsity using extremal expander graphs (Ramanujan graphs) to generate a mask which can be applied on the low-rank A,B matrices.

**Strengths:**

Using Ramanujan graphs for LoRA is interesting and empirical results show that comparable performance can be achieved with a small amount of trainable parameters.

**Weaknesses:**

- There is no comparison with other LoRA variants. In particular, approaches which enforce sparsity such as VeRA [1], RoSA [2], SoRA [3], AdaLoRA [4]
- Only RoBERTa base (125M) and RoBERTa large (355M) are evaluated. These models are small in size and there is no practical need to reduce their number of parameters when training. Experiments at a larger scale e.g., Llama 7B should be carried out.

[1] Kopiczko, D. J., Blankevoort, T., & Asano, Y. M. (2023). Vera: Vector-based random matrix adaptation. arXiv preprint arXiv:2310.11454.

[2] Nikdan, M., Tabesh, S., Crnčević, E., & Alistarh, D. (2024). Rosa: Accurate parameter-efficient fine-tuning via robust adaptation. arXiv preprint arXiv:2401.04679.

[3] Ding, N., Lv, X., Wang, Q., Chen, Y., Zhou, B., Liu, Z., & Sun, M. (2023). Sparse low-rank adaptation of pre-trained language models. arXiv preprint arXiv:2311.11696.

[4] Zhang, Q., Chen, M., Bukharin, A., Karampatziakis, N., He, P., Cheng, Y., ... & Zhao, T. (2023). AdaLoRA: Adaptive budget allocation for parameter-efficient fine-tuning. arXiv preprint arXiv:2303.10512.

**Questions:**

- How does the method differ from other sparsity-inducing LoRA methods in terms of performance and complexity?
- What is the time complexity to generate the expander masks in section 4.1?
- How are the masks updated during training? Will the updates result in masks that do not satisfy the Ramanujan criteria?

---

> ### Author Response · Authors · 2024-11-18
> **Response to Reviewer yii8**
>
> We thank the reviewer for acknowledging our work and for the constructive comments and suggestions on the manuscript.
>
> Response to Weaknesses:
>
> - We have now added comparisons with Vera [1] in Tables 2,3. Further comparisons can be found in sections 5.2.3. and 5.2.4. We are conducting additional experiments on GPT 2 and ViT architectures.
> - We have now added the new subsection 5.2.3 INSTRUCTION TUNING where the results of the experiments on Llama 2 7B have been incorporated.
>
> Response to Questions:
> - Structured Sparsity: Methods such as RoSA and AdaLoRA focus on adaptive or regularized sparsity, XoRA enforces predetermined structured sparsity via expander graphs. This ensures consistent connectivity and robustness. While methods like AdaLoRA adaptively allocate budget during training, XoRA leverages precomputed expander masks. This reduces runtime overhead associated with dynamic sparsity adaptation. Our experiments demonstrate that XoRA maintains competitive performance at higher sparsity levels compared to the other methods.
> - The computational complexity for generating the Ramanujan graph of dimension M by N using both the techniques (random bi-partite matching or deterministic via Ramanujan $r$-coverings) is $O(nMN)$ where $(n,m)$ is the bi-regularity. The tradeoff is negligible because even for large language models $M$ is typically of the order of $10^4$ and $N$ is the LoRA rank  and even with high Lora rank say $r = 128, 256, 512$  we can obtain sparse masks of low bi-regularity within a few seconds. The subsection 4.1.1 COMPLEXITY OF MASK GENERATION has been added to discuss this.
> - In XoRA, the expander masks remain static throughout training. This design choice ensures that the masks retain the Ramanujan properties and the associated spectral guarantees.
>
> [1] Dawid Jan Kopiczko, Tijmen Blankevoort, and Yuki M Asano. Vera: Vector-based random matrix adaptation. In International Conference on Learning Representations, 2023.
>
> *Updated only Table numbers from previous reply.

---

> > ### Author Response · Authors · 2024-11-24
> > **Response to Reviewer yii8 (continued)**
> >
> > Continuing on from the previous reply, in this revision, we have added further additional experimental results on DeBERTa-V3-base, GPT-2 Medium to compare with other state of the art adaptation methods like SoRA, DyLoRA, AdaLora, VeRA (sec 5.2.5 Additional Experiments). The details are presented below:
> >
> > Comparison of XoRA with existing parameter-efficient fine-tuning methods on GLUE benchmark tasks using DeBERTa-V3-base as the backbone model (top values are in bold while second best values are in italics). See Table 8 in the paper.
> > | **Method**    | **#Params** | **MRPC**                     | **STS-B**                    | **RTE**                      | **CoLA**                     | **SST-2**                    | **Avg.** |
> > |---------------|-------------|------------------------------|------------------------------|------------------------------|------------------------------|------------------------------|----------|
> > | Fine-Tune     | 184M        | 89.22 ± 0.69                | 91.59 ± 0.47                | 82.49 ± 1.48                | 69.21 ± 2.24                | 95.64 ± 0.52                | 85.63    |
> > | Adapter       | 1.41M       | 89.90 ± 2.10                | _92.21 ± 0.33_              | 82.44 ± 1.74                | 69.00 ± 0.91                | 95.16 ± 0.46                | 85.74    |
> > | Bitfit        | 0.1M        | 87.16 ± 0.58                | 89.71 ± 0.58                | 76.12 ± 1.54                | 68.70 ± 1.85                | 94.38 ± 0.28                | 83.21    |
> > | LoRA          | 1.33M       | 89.71 ± 1.32                | 91.86 ± 0.29                | 85.32 ± 0.86                | 69.73 ± 1.42                | 95.57 ± 0.21                | 86.44    |
> > | AdaLoRA       | 1.27M       | 90.22 ± 0.40                | 91.39 ± 0.25                | 87.36 ± 0.30                | 70.86 ± 1.43                | **95.95 ± 0.37**            | 87.16    |
> > | SoRA          | 0.91M       | _91.98 ± 1.16_              | **92.22 ± 0.24**            | _87.77 ± 1.56_              | _71.48 ± 1.17_              | 95.64 ± 0.23                | 87.82    |
> > | **XoRA**      | **0.33M**   | **92.02 ± 1.22**            | _92.21 ± 0.18_              | **87.91 ± 1.07**            | **71.99 ± 1.25**            | _95.92 ± 0.29_              | **88.01** |
> >
> > ---
> > ---
> >
> > Comparison of different adaptations methods for GPT-2 Medium on the E2E NLG Challenge benchmark (top values are in bold while second best values are in italics) See Table 9 in the paper.
> > | **Method**         | **#Params** | **BLEU**       | **NIST**       | **METEOR**     | **ROUGE-L**     | **CIDEr**      |
> > |---------------------|-------------|----------------|----------------|----------------|----------------|----------------|
> > | GPT2-M FT$^\dagger$ | 354.92M    | 68.2           | 8.62           | 46.2           | 71.0           | 2.47           |
> > | AdptL$^\dagger$    | 0.37M       | 66.3           | 8.41           | 45.0           | 69.8           | 2.40           |
> > | AdptL$^\dagger$    | 11.09M      | 68.9           | 8.71           | 46.1           | 71.3           | 2.47           |
> > | AdptH$^\dagger$    | 11.09M      | 67.3           | 8.50           | 46.0           | 70.7           | 2.44           |
> > | FTTop2$^\dagger$   | 25.19M      | 68.1           | 8.59           | 46.0           | 70.8           | 2.41           |
> > | PreLayer$^\dagger$ | 0.35M       | 69.7           | 8.81           | 46.1           | 71.4           | 2.49           |
> > | LoRA$^\dagger$     | 0.35M       | **70.4**       | 8.85           | **46.8**       | **71.8**       | **2.53**       |
> > | DyLoRA$^\ddagger$  | 0.39M       | 69.2           | 8.75           | 46.3           | 70.8           | 2.46           |
> > | AdaLoRA$^\star$    | 0.38M       | 68.2           | 8.58           | 44.1           | 70.7           | 2.35           |
> > | VeRA$^\star$       | 0.098M      | 70.1           | 8.81           | _46.6_         | 71.5           | 2.50           |
> > | **XoRA**           | **0.196M**  | **70.4**       | **8.86**       | 46.4           | _71.7_         | _2.51_         |

---

> > > ### Author Response · Authors · 2024-11-29
> > >
> > > Dear Reviewer,
> > > We appreciate your constructive review and have made efforts to address your concerns by reporting further experimental results and other details. We are looking forward to your feedback on the revised version.

---

> > > > ### Author Response · Authors · 2024-12-02
> > > >
> > > > Dear Reviewer yii8,
> > > >
> > > > With the extended author-reviewer discussion period ending in a few hours (Dec 2, 23:59:59 AOE), we kindly note that we have not yet received any feedback or engagement on the revised manuscript. We remain available to address any concerns or questions and request a review of the manuscript. Thank you for your time and consideration.

---

### Official Review · Reviewer_C2WH · 2024-11-03

**Soundness:** 3
**Presentation:** 2
**Contribution:** 2
**Rating:** 5
**Confidence:** 4

**Summary:**

This authors propose XLoRA which is a PEFT technique that apply sparsity to LoRA low-rank matrices through expander graph-based masking, specifically leveraging Ramanujan graphs for sparsity while maintaining connectivity. The method aims to further reduce parameter count of LoRA fine-tuning.

**Strengths:**

1. Introducing expander graph-based masking for structural sparsity in fine-tuning models is innovative.
2. The paper clearly motivates the need for reducing parameters in LoRA and the potential benefits of sparsity.

**Weaknesses:**

1. Insufficient introduction of other advanced PEFT methods. The authors mainly focus on the introduction of LoRA-related PEFT methods. While other PEFT methods, such as IA3, OFT/BOFT and prompt-based methods are not discussed.
2. Insufficient comparison with advanced PEFT methods mentioned in weakness 1.
3. The manuscript highlights the benefits of using expander graphs, but it does not discuss any potential downsides. For example, generating Ramanujan graphs can be computationally intensive, especially for very high degrees. A discussion on computational overhead and any trade-offs in performance versus simplicity of other masking techniques would be beneficial.
4. Although parameter efficiency is demonstrated, it’s not clear how XoRA impacts memory usage and computational cost during.

**Questions:**

The paper shows that XoRA can perform well even at 75% sparsity. However, is there a threshold where the sparsity level starts to significantly degrade performance? What is the relationship between that sparsity level and the rank of LoRA adapters?

---

> ### Author Response · Authors · 2024-11-18
> **Response to Reviewer C2WH**
>
> We thank the reviewer for acknowledging our work and for the constructive comments and suggestions on the manuscript.
>
> Response to Weaknesses:
>
> - A discussion on other PEFT related methods such as IA3, OFT/BOFT and prompt-based methods have now been incorporated in the end of section 2 RELATED WORK.
> - OFT/BOFT methods are mainly applied for vision finetuning tasks, in this work, we mainly concentrated on NLP tasks to demonstrate the effectiveness of our method. We are conducting additional experiments on ViT.
> -  We have added a subsection  4.1.1 COMPLEXITY OF MASK GENERATION regarding the computational complexity and also the implications for XoRA.  The tradeoff is negligible, typically, even with high Lora rank say $r =256, 512$ etc. we can obtain sparse masks of low bi-regularity within a few seconds. Section 6 has been added to discusses the limitations.
> - In subsection 5.2.4. IMPACT OF XORA ON MEMORY USAGE AND TRAINING TIME we have now added a discussion regarding the XoRA memory usage and computational time with respect to LoRA.
>
> Response to Questions:
>
> - The construction of the Ramanujan masks fixes a lower threshold on the parameter count. This is discussed in the new section 6 LIMITATIONS AND DISCUSSIONS. The $75$% sparsity was a result of this lower bound with respect to the chosen LoRA rank ($r = 8$). For higher LoRA ranks, we can achieve higher sparsity (keeping the theoretical considerations of Ramanujan masks). For example, we have added new tables Table 4, Table 5 where for instruction tuning tasks, we operate with LoRA rank $ r = 64 $ and obtain Ramanujan mask of sparsity almost $97$%, without degradation of performance. With regards to your question regarding the relationship, the theoretical lowest threshold is $ \frac{2}{r} \times 100$% parameters remaining.
>
> *Updated only Table numbers from previous reply.

---

> > ### Author Response · Authors · 2024-11-24
> > **Response to Reviewer C2WH (continued)**
> >
> > Continuing on from the previous reply, in this revision, we have added further additional experimental results on DeBERTa-V3-base, GPT-2 Medium to compare with other state of the art adaptation methods like SoRA, DyLoRA, AdaLora, VeRA (sec 5.2.5 Additional Experiments). The details are presented below:
> >
> > Comparison of XoRA with existing parameter-efficient fine-tuning methods on GLUE benchmark tasks using DeBERTa-V3-base as the backbone model (top values are in bold while second best values are in italics). See Table 8 in the paper.
> > | **Method**    | **#Params** | **MRPC**                     | **STS-B**                    | **RTE**                      | **CoLA**                     | **SST-2**                    | **Avg.** |
> > |---------------|-------------|------------------------------|------------------------------|------------------------------|------------------------------|------------------------------|----------|
> > | Fine-Tune     | 184M        | 89.22 ± 0.69                | 91.59 ± 0.47                | 82.49 ± 1.48                | 69.21 ± 2.24                | 95.64 ± 0.52                | 85.63    |
> > | Adapter       | 1.41M       | 89.90 ± 2.10                | _92.21 ± 0.33_              | 82.44 ± 1.74                | 69.00 ± 0.91                | 95.16 ± 0.46                | 85.74    |
> > | Bitfit        | 0.1M        | 87.16 ± 0.58                | 89.71 ± 0.58                | 76.12 ± 1.54                | 68.70 ± 1.85                | 94.38 ± 0.28                | 83.21    |
> > | LoRA          | 1.33M       | 89.71 ± 1.32                | 91.86 ± 0.29                | 85.32 ± 0.86                | 69.73 ± 1.42                | 95.57 ± 0.21                | 86.44    |
> > | AdaLoRA       | 1.27M       | 90.22 ± 0.40                | 91.39 ± 0.25                | 87.36 ± 0.30                | 70.86 ± 1.43                | **95.95 ± 0.37**            | 87.16    |
> > | SoRA          | 0.91M       | _91.98 ± 1.16_              | **92.22 ± 0.24**            | _87.77 ± 1.56_              | _71.48 ± 1.17_              | 95.64 ± 0.23                | 87.82    |
> > | **XoRA**      | **0.33M**   | **92.02 ± 1.22**            | _92.21 ± 0.18_              | **87.91 ± 1.07**            | **71.99 ± 1.25**            | _95.92 ± 0.29_              | **88.01** |
> >
> > ---
> > ---
> >
> > Comparison of different adaptations methods for GPT-2 Medium on the E2E NLG Challenge benchmark (top values are in bold while second best values are in italics) See Table 9 in the paper.
> > | **Method**         | **#Params** | **BLEU**       | **NIST**       | **METEOR**     | **ROUGE-L**     | **CIDEr**      |
> > |---------------------|-------------|----------------|----------------|----------------|----------------|----------------|
> > | GPT2-M FT$^\dagger$ | 354.92M    | 68.2           | 8.62           | 46.2           | 71.0           | 2.47           |
> > | AdptL$^\dagger$    | 0.37M       | 66.3           | 8.41           | 45.0           | 69.8           | 2.40           |
> > | AdptL$^\dagger$    | 11.09M      | 68.9           | 8.71           | 46.1           | 71.3           | 2.47           |
> > | AdptH$^\dagger$    | 11.09M      | 67.3           | 8.50           | 46.0           | 70.7           | 2.44           |
> > | FTTop2$^\dagger$   | 25.19M      | 68.1           | 8.59           | 46.0           | 70.8           | 2.41           |
> > | PreLayer$^\dagger$ | 0.35M       | 69.7           | 8.81           | 46.1           | 71.4           | 2.49           |
> > | LoRA$^\dagger$     | 0.35M       | **70.4**       | 8.85           | **46.8**       | **71.8**       | **2.53**       |
> > | DyLoRA$^\ddagger$  | 0.39M       | 69.2           | 8.75           | 46.3           | 70.8           | 2.46           |
> > | AdaLoRA$^\star$    | 0.38M       | 68.2           | 8.58           | 44.1           | 70.7           | 2.35           |
> > | VeRA$^\star$       | 0.098M      | 70.1           | 8.81           | _46.6_         | 71.5           | 2.50           |
> > | **XoRA**           | **0.196M**  | **70.4**       | **8.86**       | 46.4           | _71.7_         | _2.51_         |

---

> > > ### Author Response · Authors · 2024-11-29
> > >
> > > Dear Reviewer,
> > > We appreciate your constructive review and have made efforts to address your concerns by reporting further experimental results and other details. We are looking forward to your feedback on the revised version.

---

> > > > ### Author Response · Authors · 2024-12-02
> > > >
> > > > Dear Reviewer C2WH,
> > > >
> > > > With the extended author-reviewer discussion period ending in a few hours (Dec 2, 23:59:59 AOE), we kindly note that we have not yet received any feedback or engagement on the revised manuscript. We remain available to address any concerns or questions and request a review of the manuscript. Thank you for your time and consideration.

---

### Official Review · Reviewer_XhuP · 2024-11-03

**Soundness:** 3
**Presentation:** 3
**Contribution:** 2
**Rating:** 6
**Confidence:** 3

**Summary:**

In this paper, the authors propose a novel LoRA-based optimization framework centered around Expander Graphs, specifically Ramanujan graphs. Ramanujan graphs possess remarkable properties that maintain high edge connectivity even at significant sparsity levels. This characteristic provides an opportunity to further sparsify the original LoRA while preserving comparable performance. Experiments conducted on the GLUE benchmark using the RoBERTa model demonstrate that, at approximately 50% sparsity in comparison to LoRA, the proposed method exhibits similar performance with LoRA.

**Strengths:**

1. The theory of Ramanujan graphs offers a formal mathematical framework for analyzing the sparsity of LoRA. This could represent a novel perspective for future LoRA optimization and, furthermore, opens avenues for mathematical analysis applicable to general pruning algorithms.
2. The presentation is commendable overall; the writing is clear and cohesive, making it easy to comprehend.

**Weaknesses:**

1. The novelty of this paper appears to be somewhat ambiguous. While the theory of Ramanujan graphs is well-established, the paper allocates considerable space to introducing the background. It remains unclear what new contributions this paper presents, as well as the distinction between existing knowledge and the authors' original work. I believe that merely applying an established technique to a new problem does not sufficiently meet the standards for publication at ICLR.
2. The experimental results do not sufficiently demonstrate the superiority of the proposed method in several respects:
2.1. The numerical results are not particularly striking. With 50% sparsity, XoRA is comparable to the original LoRA, while further increases in sparsity lead to diminished results. It is unclear how to effectively balance sparsity and performance.
2.2 The experiments are somewhat limited. I would recommend expanding the evaluation beyond the RoBERTa model to include BERT, GPT, and LLaMA. These well-known language models, along with multimodal models, could provide a more comprehensive perspective for assessing XoRA. In addition to the GLUE benchmark, I suggest incorporating the WikiText and PTB datasets for language model evaluation and MMLU for multimodal model assessment.
2.3. Furthermore, I propose including comparisons with methods such as LoRAPrune and LLM-pruner, as both approaches address the problem from a sparsity perspective.
2.4 Regarding practical applications, sparsity metrics may not be reliable indicators. It would be beneficial to see actual runtime improvements achieved with XoRA, such as inference latency and training costs. These critical aspects are currently omitted from the paper.

**Questions:**

In light of the identified weaknesses, I have the following inquiries:

1. What is the main contribution of this paper?
2. Could you provide additional insights into XoRA? Specifically, what type of sparsity pattern does the method have, and how might this impact runtime performance, including inference latency and training costs?
3. How can one effectively balance sparsity and performance with XoRA?

---

> ### Author Response · Authors · 2024-11-18
> **Response to Reviewer XhuP**
>
> We thank the reviewer for acknowledging our work and for the constructive comments and suggestions on the manuscript.
>
> Response to Weaknesses:
>
>   - We have added a Contributions section (sec 1.1). The discussion on Ramanujan graphs has been moved to the appendix. The use of Ramanujan graph based sparsification for LoRA finetuning is novel and identifies winning lottery tickets in these matrices.
>   -  In the new subsection 5.2.3 INSTRUCTION TUNING we present experimental results for the Llama 2 7B model. The sparsity presented before depended on the LoRA rank. We present higher sparsity results for Instruction tuning (see Table 4). Figure 2 and Table 5 demonstrates the effectiveness of the method at higher sparsity levels for different tasks. Table 7 does a memory usage and training time comparison. The method manages to reduce both with respect to LoRA (a discussion regarding this quantity wrt other state of the art sparsification technique is also included). Additional experiments on GPT 2 and ViT will be added.
>
> Response to Questions:
>
>   - We have now added a Contributions section (sec 1.1).
>   - For random generation we get a random adjacency matrix which satisfies the Ramanujan criteria. For deterministic generation using Cayley graphs, we obtain more structured matrices (adjacency matrices of Cayley graphs which are vertex transitive). However, even for random Ramanujan masks we obtain a reduction in memory usage and in training time (subsection 5.2.4). Also another advantage of XoRA is that the matrices after masking preserve the rank. So for finetuning tasks where a high LoRA rank is needed, methods which reduces the LoRA rank may be at a disadvantage.
>   - Table 4 gives a comparison of sparsity and performance for Instruction tuning tasks. Also a section on limitations has now been added (section 6).
>
> *Updated only Table numbers from previous reply.

---

> > ### Author Response · Authors · 2024-11-24
> > **Response to Reviewer XhuP (continued)**
> >
> > Continuing on from the previous reply, in this revision, we have added further additional experimental results on DeBERTa-V3-base, GPT-2 Medium to compare with other state of the art adaptation methods like SoRA, DyLoRA, AdaLora, VeRA (sec 5.2.5 Additional Experiments). The details are presented below:
> >
> > Comparison of XoRA with existing parameter-efficient fine-tuning methods on GLUE benchmark tasks using DeBERTa-V3-base as the backbone model (top values are in bold while second best values are in italics). See Table 8 in the paper.
> > | **Method**    | **#Params** | **MRPC**                     | **STS-B**                    | **RTE**                      | **CoLA**                     | **SST-2**                    | **Avg.** |
> > |---------------|-------------|------------------------------|------------------------------|------------------------------|------------------------------|------------------------------|----------|
> > | Fine-Tune     | 184M        | 89.22 ± 0.69                | 91.59 ± 0.47                | 82.49 ± 1.48                | 69.21 ± 2.24                | 95.64 ± 0.52                | 85.63    |
> > | Adapter       | 1.41M       | 89.90 ± 2.10                | _92.21 ± 0.33_              | 82.44 ± 1.74                | 69.00 ± 0.91                | 95.16 ± 0.46                | 85.74    |
> > | Bitfit        | 0.1M        | 87.16 ± 0.58                | 89.71 ± 0.58                | 76.12 ± 1.54                | 68.70 ± 1.85                | 94.38 ± 0.28                | 83.21    |
> > | LoRA          | 1.33M       | 89.71 ± 1.32                | 91.86 ± 0.29                | 85.32 ± 0.86                | 69.73 ± 1.42                | 95.57 ± 0.21                | 86.44    |
> > | AdaLoRA       | 1.27M       | 90.22 ± 0.40                | 91.39 ± 0.25                | 87.36 ± 0.30                | 70.86 ± 1.43                | **95.95 ± 0.37**            | 87.16    |
> > | SoRA          | 0.91M       | _91.98 ± 1.16_              | **92.22 ± 0.24**            | _87.77 ± 1.56_              | _71.48 ± 1.17_              | 95.64 ± 0.23                | 87.82    |
> > | **XoRA**      | **0.33M**   | **92.02 ± 1.22**            | _92.21 ± 0.18_              | **87.91 ± 1.07**            | **71.99 ± 1.25**            | _95.92 ± 0.29_              | **88.01** |
> >
> > ---
> > ---
> >
> > Comparison of different adaptations methods for GPT-2 Medium on the E2E NLG Challenge benchmark (top values are in bold while second best values are in italics) See Table 9 in the paper.
> > | **Method**         | **#Params** | **BLEU**       | **NIST**       | **METEOR**     | **ROUGE-L**     | **CIDEr**      |
> > |---------------------|-------------|----------------|----------------|----------------|----------------|----------------|
> > | GPT2-M FT$^\dagger$ | 354.92M    | 68.2           | 8.62           | 46.2           | 71.0           | 2.47           |
> > | AdptL$^\dagger$    | 0.37M       | 66.3           | 8.41           | 45.0           | 69.8           | 2.40           |
> > | AdptL$^\dagger$    | 11.09M      | 68.9           | 8.71           | 46.1           | 71.3           | 2.47           |
> > | AdptH$^\dagger$    | 11.09M      | 67.3           | 8.50           | 46.0           | 70.7           | 2.44           |
> > | FTTop2$^\dagger$   | 25.19M      | 68.1           | 8.59           | 46.0           | 70.8           | 2.41           |
> > | PreLayer$^\dagger$ | 0.35M       | 69.7           | 8.81           | 46.1           | 71.4           | 2.49           |
> > | LoRA$^\dagger$     | 0.35M       | **70.4**       | 8.85           | **46.8**       | **71.8**       | **2.53**       |
> > | DyLoRA$^\ddagger$  | 0.39M       | 69.2           | 8.75           | 46.3           | 70.8           | 2.46           |
> > | AdaLoRA$^\star$    | 0.38M       | 68.2           | 8.58           | 44.1           | 70.7           | 2.35           |
> > | VeRA$^\star$       | 0.098M      | 70.1           | 8.81           | _46.6_         | 71.5           | 2.50           |
> > | **XoRA**           | **0.196M**  | **70.4**       | **8.86**       | 46.4           | _71.7_         | _2.51_         |

---

> > > ### Comment · Reviewer_XhuP · 2024-11-25
> > >
> > > I would like to express my gratitude to the authors for the comprehensive updates on the experiments and their further elaboration on the paper's contributions. I believe that some of my concerns have been addressed, and we now have a clearer understanding of the proposed method.
> > >
> > > 1. As I had anticipated, the reduction in GPU memory usage and training time, as shown in Table 7, is minimal. This is unfortunate, as it could significantly limit the practical applicability of XoRA. I would be particularly interested in seeing data on inference time reduction, as this is the most relevant factor for real-world deployment. I hope the authors will share their findings on this matter.
> > >
> > > 2. Apart from GPU memory and training time, the performance of XoRA and VeRA appears to be comparable, at least. From Tables 5 and 6, I cannot identify a fair comparison in terms of computational resources, making it difficult to determine which is superior. When evaluating the GLUE benchmark, VeRA seems to outperform XoRA given RoBERTa base and RoBERTa large models.
> > >
> > > 3. The most intriguing observation arises from Table 4, which shows that XoRA consistently surpasses the original LoRA. This finding is somewhat inconsistent with Table 1, where LoRA appears slightly more advantageous. Is there something specific about instruction tuning that aligns particularly well with XoRA? I would appreciate if the authors could explore this aspect further.
> > >
> > > As for now, I decide to uphold my original score.

---

> > > > ### Author Response · Authors · 2024-11-26
> > > > **Response to Reviewer XhuP**
> > > >
> > > > We thank the reviewer for their valuable feedback and insightful comments. Below, we address the concerns raised, focusing on highlighting XoRA’s efficiency and advantages across the key dimensions of parameter efficiency, computational performance, and applicability to real-world deployment.
> > > >
> > > > ---
> > > >
> > > > ### 1. Computational Requirements, Inference Overhead
> > > >
> > > > We acknowledge the reviewer's observation about the GPU memory and training time reduction in Table 7. While these reductions may appear modest and minimal in isolation in Table 7, to the best of our knowledge, the reductions XoRA demonstrates are significant enough as foundational models continue to scale. Most memory usage stems from loading the base model, which is required regardless of the fine-tuning method. XoRA achieves savings by reducing the memory required for gradients and optimizer states associated with the adapter parameters with structured sparsity. While this is impactful, it is naturally less pronounced because of the dominance of base model memory and activation memory required, which is mostly invariant in any fine-tuning method. While the percentage reductions may seem modest, the absolute memory and time savings scale linearly with larger models and higher ranks.
> > > >
> > > > **No Additional Inference Overhead**: XoRA's adapter matrices \( A \) and \( B \) are structurally identical to LoRA, with updates constrained to the sparse positions dictated by the expander masks. These masks are only required during training and are merged into the model weights for inference, preserving efficiency. This is the same for VeRA adapters as well; however, in VeRA, the scaling vectors introduce additional computational overhead (multiplication with scaling factors). In contrast, XoRA avoids this computational cost.
> > > >
> > > > The reductions in GPU memory usage and training time demonstrated by XoRA are substantial when considering real-world deployment, especially for large foundational models. In real-world scenarios, this is particularly critical when managing hundreds of specialized adapters across large foundational models for personalization, task switching, and domain-specific fine-tuning. This context amplifies XoRA's value proposition. While any reparameterized PEFT method involves matrix multiplication bottlenecks, XoRA’s structured sparsity optimizations can further minimize these overheads effectively.
> > > >
> > > > ---
> > > >
> > > > ### 2. Reported Results and Variability
> > > >
> > > > The values reported in RoBERTa base and RoBERTa large NLU experiments are the **median** of five random seeds, with standard deviations provided to reflect variability. For certain seeds, XoRA exceeds the accuracy of both LoRA and VeRA. XoRA consistently maintains or exceeds LoRA performance across GLUE tasks while only using ¼ of the parameters for a rank-8 configuration, which is the maximum sparsity level from expander-based structured sparsity.
> > > >
> > > > The reported accuracy of VeRA (and all baselines) is taken from the prior reported values, as mentioned in the caption of Table 3. (*For VeRA, we report the scores published in [Kopiczko et al., 2023]. Our replication experiments yielded slightly lower averages: 83.0 for RoBERTa-base and 86.2 for RoBERTa-large due to seeds and variations in hyperparameters.)
> > > >
> > > > ---
> > > >
> > > > ### 3. Structured Sparsity and Instruction Tuning
> > > >
> > > > Unlike LoRA, which often introduces noise into weight updates, XoRA’s structured sparsity introduces an inherent regularization to the weight updates, reducing noise and improving task-specific adaptation. This is particularly impactful in instruction tuning, where targeted updates are crucial for aligning the model with task requirements.
> > > >
> > > > The results highlight that LoRA updates often include redundancy, whereas XoRA’s expander-mask-based constraints focus updates on the most impactful parameters. This aligns well with instruction tuning, where targeted updates lead to better alignment and generalization. We have also quantitatively assessed and compared factual accuracy and knowledge retention of LoRA and XoRA instruction fine-tuned models.
> > > >
> > > > To further validate XoRA’s advantages, we have included additional results in the revised version of the paper, evaluating XoRA on:
> > > > - **GPT-2 Medium** (Natural Language Generation Tasks)
> > > > - **DeBERTa V3 Base** models.
> > > >
> > > > These results are presented in **Tables 8 and 9** of the revised version.

---

> > > > > ### Comment · Reviewer_XhuP · 2024-11-26
> > > > >
> > > > > I appreciate the authors' prompt response. Regarding inference overhead, I anticipated a reduction in inference runtime due to the increased sparsity introduced by XoRA. It would be somewhat surprising if there were no benefits to inference efficiency.
> > > > >
> > > > > With respect to XoRA's performance on instruction tuning, it is evident that XoRA performs well in instruction tuning tasks; however, I did not find any particularly insightful information in the response. If there is indeed a correlation between structured sparsity and instruction tuning, it could represent a significant finding for this paper.

---

> > > > > > ### Author Response · Authors · 2024-11-29
> > > > > >
> > > > > > 1. The reduction in peak memory usage from LoRA was 9.2% (for Llama 2 7B) and in training time was 3.2%. This is without kernel optimisation techniques. Our focus for this work was mainly on reduction in number of parameters. We believe that kernel optimisation techniques will help in further reduction. In a future work we propose to pursue this.
> > > > > >
> > > > > > 2. Regarding the inference time, since only gradient updates are reduced - the inference time is not reduced. This is because the optimization primarily targets the gradient updates during training, leaving the full parameter matrices intact for inference.
> > > > > >
> > > > > > 3. We are doing further experiments to understand these correlation in detail.

---

> > > > > > > ### Comment · Reviewer_XhuP · 2024-11-30
> > > > > > >
> > > > > > > I would like to thank the authors for their further clarification. Although it is unfortunate that the improvements in memory usage, training time, and inference cost are modest, I still regard this paper as a solid contribution in its current form.
> > > > > > >
> > > > > > > Furthermore, I encourage the authors to investigate the correlation between structured sparsity and instruction tuning, as this exploration could serve as a significant highlight of the paper. Otherwise, it might be yet another publication in ICLR.
> > > > > > >
> > > > > > > Considering the comprehensiveness and rigor of the work, I decide to raise my score. However, I look forward to seeing the results related to structured sparsity and instruction tuning.

---

> > > > > > > > ### Author Response · Authors · 2024-12-01
> > > > > > > >
> > > > > > > > We appreciate greatly your constructive feedback and acknowledgement of our work. Thank you. Yes, as follow-up, we have started working on the interesting direction that you suggested.

---

### Official Review · Reviewer_k7nD · 2024-11-03

**Soundness:** 2
**Presentation:** 2
**Contribution:** 1
**Rating:** 3
**Confidence:** 5

**Summary:**

The paper proposes using expanders for employing sparsity on LoRA parameters.

The proposed method, XoRA, was examined on GLUE benchmarks with Roberta models.

**Strengths:**

Expanders have been used in deep learning for employing sparsity constraints on network architectures.

Their application for LoRA is interesting and the initial results are promising.

**Weaknesses:**

Although the initial results are promising, the experimental analyses are limited. Therefore, the contribution of the work is not clear.

More precisely, the proposed XoRA should be compared with the state-of-the-art sparse LoRA methods in different tasks using various models to show its merit and superiority.

**Questions:**

- There are some problems with the notation. For instance, the gradient update was defined by both AxB and BA. If BA is matrix multiplication of A and B, then, what does x denote?

- What does $h_V(\Gamma)$ denote?

- How does the XoRA perform with different LLMs and VLMs on additional tasks such as text and image generation?

---

> ### Author Response · Authors · 2024-11-18
> **Response to Reviewer k7nD**
>
> We thank the reviewer for acknowledging our work and for the constructive comments and suggestions on the manuscript.
>
> Response to Weaknesses:
>
>   - We have now added additional comparisons of XoRA with other peft adaptation methods especially VeRA [1]. These are detailed in subsections 5.2.2., 5.2.3, 5.2.4. Additional experiments on GPT 2 and ViT will be added.
>   - We list the Contributions in section 1.1
>
> Response to Questions:
>
>   - This was a typo in the introduction. It has now been corrected.
>   - $\mathbf{h}_{V}(\Gamma)$ denotes the vertex Cheeger constant. The typo in the definition has now been corrected.
>   - In subsection 5.2.3. INSTRUCTION TUNING, we have included detailed analyses of how XoRA performs with Llama 2 7B and additional experiments on GPT 2 and ViT will be added.
>
> [1] Dawid Jan Kopiczko, Tijmen Blankevoort, and Yuki M Asano. Vera: Vector-based random matrix
> adaptation. In International Conference on Learning Representations, 2023.

---

> > ### Author Response · Authors · 2024-11-24
> > **Response to Reviewer k7nD (continued)**
> >
> > Continuing on from the previous reply, in this revision, we have added further additional experimental results on DeBERTa-V3-base, GPT-2 Medium to compare with other state of the art adaptation methods like SoRA, DyLoRA, AdaLora, VeRA (sec 5.2.5 Additional Experiments). The details are presented below:
> >
> > Comparison of XoRA with existing parameter-efficient fine-tuning methods on GLUE benchmark tasks using DeBERTa-V3-base as the backbone model (top values are in bold while second best values are in italics). See Table 8 in the paper.
> > | **Method**    | **#Params** | **MRPC**                     | **STS-B**                    | **RTE**                      | **CoLA**                     | **SST-2**                    | **Avg.** |
> > |---------------|-------------|------------------------------|------------------------------|------------------------------|------------------------------|------------------------------|----------|
> > | Fine-Tune     | 184M        | 89.22 ± 0.69                | 91.59 ± 0.47                | 82.49 ± 1.48                | 69.21 ± 2.24                | 95.64 ± 0.52                | 85.63    |
> > | Adapter       | 1.41M       | 89.90 ± 2.10                | _92.21 ± 0.33_              | 82.44 ± 1.74                | 69.00 ± 0.91                | 95.16 ± 0.46                | 85.74    |
> > | Bitfit        | 0.1M        | 87.16 ± 0.58                | 89.71 ± 0.58                | 76.12 ± 1.54                | 68.70 ± 1.85                | 94.38 ± 0.28                | 83.21    |
> > | LoRA          | 1.33M       | 89.71 ± 1.32                | 91.86 ± 0.29                | 85.32 ± 0.86                | 69.73 ± 1.42                | 95.57 ± 0.21                | 86.44    |
> > | AdaLoRA       | 1.27M       | 90.22 ± 0.40                | 91.39 ± 0.25                | 87.36 ± 0.30                | 70.86 ± 1.43                | **95.95 ± 0.37**            | 87.16    |
> > | SoRA          | 0.91M       | _91.98 ± 1.16_              | **92.22 ± 0.24**            | _87.77 ± 1.56_              | _71.48 ± 1.17_              | 95.64 ± 0.23                | 87.82    |
> > | **XoRA**      | **0.33M**   | **92.02 ± 1.22**            | _92.21 ± 0.18_              | **87.91 ± 1.07**            | **71.99 ± 1.25**            | _95.92 ± 0.29_              | **88.01** |
> >
> > ---
> > ---
> >
> > Comparison of different adaptations methods for GPT-2 Medium on the E2E NLG Challenge benchmark (top values are in bold while second best values are in italics) See Table 9 in the paper.
> > | **Method**         | **#Params** | **BLEU**       | **NIST**       | **METEOR**     | **ROUGE-L**     | **CIDEr**      |
> > |---------------------|-------------|----------------|----------------|----------------|----------------|----------------|
> > | GPT2-M FT$^\dagger$ | 354.92M    | 68.2           | 8.62           | 46.2           | 71.0           | 2.47           |
> > | AdptL$^\dagger$    | 0.37M       | 66.3           | 8.41           | 45.0           | 69.8           | 2.40           |
> > | AdptL$^\dagger$    | 11.09M      | 68.9           | 8.71           | 46.1           | 71.3           | 2.47           |
> > | AdptH$^\dagger$    | 11.09M      | 67.3           | 8.50           | 46.0           | 70.7           | 2.44           |
> > | FTTop2$^\dagger$   | 25.19M      | 68.1           | 8.59           | 46.0           | 70.8           | 2.41           |
> > | PreLayer$^\dagger$ | 0.35M       | 69.7           | 8.81           | 46.1           | 71.4           | 2.49           |
> > | LoRA$^\dagger$     | 0.35M       | **70.4**       | 8.85           | **46.8**       | **71.8**       | **2.53**       |
> > | DyLoRA$^\ddagger$  | 0.39M       | 69.2           | 8.75           | 46.3           | 70.8           | 2.46           |
> > | AdaLoRA$^\star$    | 0.38M       | 68.2           | 8.58           | 44.1           | 70.7           | 2.35           |
> > | VeRA$^\star$       | 0.098M      | 70.1           | 8.81           | _46.6_         | 71.5           | 2.50           |
> > | **XoRA**           | **0.196M**  | **70.4**       | **8.86**       | 46.4           | _71.7_         | _2.51_         |

---

> > > ### Author Response · Authors · 2024-11-29
> > >
> > > Dear Reviewer,
> > > We appreciate your constructive review and have made efforts to address your concerns by reporting further experimental results and other details. We are looking forward to your feedback on the revised version.

---

> ### Author Response · Authors · 2024-12-02
>
> Dear Reviewer k7nD,
>
> With the extended author-reviewer discussion period ending in a few hours (Dec 2, 23:59:59 AOE), we kindly note that we have not yet received any feedback or engagement on the revised manuscript. We remain available to address any concerns or questions and request a review of the manuscript. Thank you for your time and consideration.

---

### Official Review · Reviewer_SBZR · 2024-11-05

**Soundness:** 3
**Presentation:** 3
**Contribution:** 4
**Rating:** 6
**Confidence:** 5

**Summary:**

The paper presents a novel parameter-efficient fine-tuning method, XoRA, which leverages Ramanujan expander graphs to sparsify the low-rank matrices used in LoRA (Low-Rank Adaptation). This interdisciplinary approach integrates graph theory with machine learning to reduce the number of trainable parameters while preserving performance, showing potential for applications where computational resources are limited. The paper’s strengths lie in its innovative methodology and solid theoretical grounding. However, further clarity on computational complexity, practical scalability, and detailed explanations of technical terms would make the work more accessible and impactful. These minor revisions would strengthen the paper’s presentation and broaden its applicability.

**Strengths:**

1. Originality: XoRA brings novelty in sparsity for fine-tuning by proposing the use of Ramanujan expander graphs-a very unique conjunction between graph theory and machine learning.
2. Quality: The theoretical grounding is solid; the empirical results support the proposed approach. Employing expander graphs is an option that adds robustness and connectivity, which are well-justified in the paper.
3. Clarity: Generally, the methodology is well-structured with a clearly described experimental setup. Subsequent sections develop well from the preceding ones, and thus it is not too hard to follow.
4. Impact: XoRA resolves one of the basic issues in parameter-efficient fine-tuning and extends the usability of LLMs in resource-constrained settings. Insights drawn through XoRA will be useful in developing efficient model adaptation in different applications.

**Weaknesses:**

1. Complexity of Graph Generation: This paper is expected to further discuss computational complexity with regard to generating an expander graph, especially in relation to large and realistic applications. This would help the audience make a judgment based on how practically feasible XoRA would be.
2. Limited Benchmark Diversity: Experimental results have been presented for only one GLUE benchmark with RoBERTa as the model backbone. In this aspect, increasing the diversity of benchmarks or model types will strengthen such results more reliably for other fine-tuning scenarios.
3. Technical Clarity: Several technical explanations, in particular about the Cheeger constant and the Ramanujan graph construction, assume some prior background in graph theory. Adding intuitive explanations for these terms could improve accessibility for a wider readership.
4. Visual Comparisons: The paper would largely benefit from more schematics, diagrams, or tables comparing performances and/or the efficiency of parameters of XoRA against the other methods of fine-tuning. That way, it would be easier for the readers to perceive the advantage of the method at once.

**Questions:**

1. Computational Feasibility: Would the authors elaborate on the computational cost of the generation and use of large-scale Ramanujan expander graphs? Is there any trade-off in using sparse operators for a gain in efficiency versus the computational expense in generating the expander masks?
2. Has the XoRA approach been theoretically evaluated or verified for architectures that extend beyond RoBERTa? The paper's impact would be significantly increased by an understanding of its adaptability across various model structures.
3. Limitations of expander graph-based sparsity: What type of models or under what conditions does this method fail to work? Knowing the limitation, if any, would give a balanced view of the scope of the method.

---

> ### Author Response · Authors · 2024-11-18
> **Response to reviewer SBZR**
>
> We thank the reviewer for acknowledging our work and for the constructive comments and suggestions on the manuscript.
>
> Response to Weaknesses:
>
> - Complexity of Graph Generation: The computational complexity for generating the Ramanujan graph of dimension \( M by N \) using both the techniques (random bi-partite matching or deterministic via Ramanujan \( r \)-coverings) is \( O(mMN) \) where \( (n, m) \) is the bi-regularity. We have added a subsection 4.1.1 COMPLEXITY OF MASK GENERATION discussing this, along with implications for XoRA.
>
> - Limited Benchmark Diversity: In the new section subsection 5.2.3 INSTRUCTION TUNING, we now present results of our experiments on the Llama 2 7B model. Table 4, Table 5 and Figure 3 provides a comparison of performance improvements of XoRA over LoRA, along with parameter sparsity. Additional experiments on GPT 2 and ViT will be added.
>
> - Technical Clarity: To improve accessibility, we have added intuitive explanations and visual aids about the Cheeger constant (see Figure 3) and properties of Ramanujan graphs (see Figure 4) in the appendix. Discussions about expanders have been moved to the appendix due to space constraints.
>
> - Visual Comparisons: We have also compared XoRA with more peft method like VeRA. See subsections 5.2.2, 5.2.3. alongwith table 7 in 5.2.4.
>
> Response to Questions:
>
> - Computational Feasibility: The computational complexity for generating the Ramanujan graph of dimension \( M \times N \) using both techniques is \( O(mMN) \). The tradeoff is negligible because even for large language models (\( M \) of the order \( 10^4 \)) and high LoRA rank \( r = 64, 128, 256 \), sparse masks of low bi-regularity can be obtained within a few seconds.
>
> - Adaptability Beyond RoBERTa: In the new subsection 5.2.3 INSTRUCTION TUNING we present experimental results for the Llama 2 7B model. Table 4,Table 5, Table 7 and Figure 2 illustrates performance improvements and memory and time improvements of XoRA over LoRA, demonstrating adaptability across different architectures and feasibility.
>
> - Limitations of Expander Graph-Based Sparsity: A discussion of the method's limitations has been added in section 6 LIMITATIONS AND DISCUSSIONS.
>
> *Updated only Table numbers from previous reply.

---

> > ### Author Response · Authors · 2024-11-24
> > **Response to reviewer SBZR (continued)**
> >
> > Continuing on from the previous reply, in this revision, we have added further additional experimental results on DeBERTa-V3-base, GPT-2 Medium to compare with other state of the art adaptation methods like SoRA, DyLoRA, AdaLora, VeRA (sec 5.2.5 Additional Experiments). The details are presented below:
> >
> > Comparison of XoRA with existing parameter-efficient fine-tuning methods on GLUE benchmark tasks using DeBERTa-V3-base as the backbone model (top values are in bold while second best values are in italics). See Table 8 in the paper.
> > | **Method**    | **#Params** | **MRPC**                     | **STS-B**                    | **RTE**                      | **CoLA**                     | **SST-2**                    | **Avg.** |
> > |---------------|-------------|------------------------------|------------------------------|------------------------------|------------------------------|------------------------------|----------|
> > | Fine-Tune     | 184M        | 89.22 ± 0.69                | 91.59 ± 0.47                | 82.49 ± 1.48                | 69.21 ± 2.24                | 95.64 ± 0.52                | 85.63    |
> > | Adapter       | 1.41M       | 89.90 ± 2.10                | _92.21 ± 0.33_              | 82.44 ± 1.74                | 69.00 ± 0.91                | 95.16 ± 0.46                | 85.74    |
> > | Bitfit        | 0.1M        | 87.16 ± 0.58                | 89.71 ± 0.58                | 76.12 ± 1.54                | 68.70 ± 1.85                | 94.38 ± 0.28                | 83.21    |
> > | LoRA          | 1.33M       | 89.71 ± 1.32                | 91.86 ± 0.29                | 85.32 ± 0.86                | 69.73 ± 1.42                | 95.57 ± 0.21                | 86.44    |
> > | AdaLoRA       | 1.27M       | 90.22 ± 0.40                | 91.39 ± 0.25                | 87.36 ± 0.30                | 70.86 ± 1.43                | **95.95 ± 0.37**            | 87.16    |
> > | SoRA          | 0.91M       | _91.98 ± 1.16_              | **92.22 ± 0.24**            | _87.77 ± 1.56_              | _71.48 ± 1.17_              | 95.64 ± 0.23                | 87.82    |
> > | **XoRA**      | **0.33M**   | **92.02 ± 1.22**            | _92.21 ± 0.18_              | **87.91 ± 1.07**            | **71.99 ± 1.25**            | _95.92 ± 0.29_              | **88.01** |
> >
> > ---
> > ---
> >
> > Comparison of different adaptations methods for GPT-2 Medium on the E2E NLG Challenge benchmark (top values are in bold while second best values are in italics) See Table 9 in the paper.
> > | **Method**         | **#Params** | **BLEU**       | **NIST**       | **METEOR**     | **ROUGE-L**     | **CIDEr**      |
> > |---------------------|-------------|----------------|----------------|----------------|----------------|----------------|
> > | GPT2-M FT$^\dagger$ | 354.92M    | 68.2           | 8.62           | 46.2           | 71.0           | 2.47           |
> > | AdptL$^\dagger$    | 0.37M       | 66.3           | 8.41           | 45.0           | 69.8           | 2.40           |
> > | AdptL$^\dagger$    | 11.09M      | 68.9           | 8.71           | 46.1           | 71.3           | 2.47           |
> > | AdptH$^\dagger$    | 11.09M      | 67.3           | 8.50           | 46.0           | 70.7           | 2.44           |
> > | FTTop2$^\dagger$   | 25.19M      | 68.1           | 8.59           | 46.0           | 70.8           | 2.41           |
> > | PreLayer$^\dagger$ | 0.35M       | 69.7           | 8.81           | 46.1           | 71.4           | 2.49           |
> > | LoRA$^\dagger$     | 0.35M       | **70.4**       | 8.85           | **46.8**       | **71.8**       | **2.53**       |
> > | DyLoRA$^\ddagger$  | 0.39M       | 69.2           | 8.75           | 46.3           | 70.8           | 2.46           |
> > | AdaLoRA$^\star$    | 0.38M       | 68.2           | 8.58           | 44.1           | 70.7           | 2.35           |
> > | VeRA$^\star$       | 0.098M      | 70.1           | 8.81           | _46.6_         | 71.5           | 2.50           |
> > | **XoRA**           | **0.196M**  | **70.4**       | **8.86**       | 46.4           | _71.7_         | _2.51_         |

---

> > ### Comment · Reviewer_SBZR · 2024-11-28
> >
> > I thank the authors for their comprehensive and constructive responses to all comments and concerns I have raised in my review. Indeed, these extensive revisions and additional experiments really strengthen the manuscript a lot.
> > It does include additional complexity analysis, which adds clarity to computational feasibility.
> > Increasing diversity of the benchmark, which includes Llama 2 7B, GPT-2 Medium, and DeBERTa-V3, contributes more to the increment in robustness.
> > It becomes more accessible to a broader audience when one includes visual aids and clarifies technical details.
> > The open discussion appearing in the new Limitations section gives valuable balance to the paper.
> > I have no further comments. The manuscript meets the standards for publication.

---

> > > ### Author Response · Authors · 2024-11-29
> > >
> > > We sincerely thank the reviewer for the thoughtful feedback and kind words regarding the improvements to our manuscript. We are
> > > grateful for your acknowledgment of the additional complexity analysis, expanded benchmarks (Llama 2 7B, GPT2-Medium, Deberta-V3-Base) and the inclusion of visual aids and other discussions. We deeply appreciate your recognition that the manuscript meets the standards for publication. Given the significant revisions and additional contributions made in response to your comments, would you be willing to reconsider your initial score? Once again, we thank you for your time and valuable insights, which have greatly improved our work.

---

### Meta-Review · Area_Chair_w6Rv · 2024-12-19

**Metareview:**

This paper mainly focuses on efficiently sparsifying LoRA matrices for finetuning. XoRA is proposed and achieves comparable performance with LoRA. After the response, it receives mixed ratings, including two rejects, one borderline reject, and two borderline accept. The advantages, including the interesting idea and good presentation, are well recognized by the reviewers. However, they are also concerned about the limited improvement, insufficient experiments, etc. I agree with the reviewers. The current manuscript does not meet the requirements of this top conference. I suggest the authors carefully revise the paper and submit it to another relevant venue.

**Additional Comments On Reviewer Discussion:**

The response well addresses part of the concerns raised by the reviewers. The Reviewer SBZR has no concerns and still gives a borderline accept rating. The reviewer XhuP increased the initial rating but still had several concerns. Other reviewers maintain their original scores. I think there is still a lot of room for improvement.

---

### Decision · Program_Chairs · 2025-01-22

Reject